# Proteomics Identifies Substrates and a Novel Component in hSnd2-Dependent ER Protein Targeting

**DOI:** 10.3390/cells11182925

**Published:** 2022-09-19

**Authors:** Andrea Tirincsi, Sarah O’Keefe, Duy Nguyen, Mark Sicking, Johanna Dudek, Friedrich Förster, Martin Jung, Drazena Hadzibeganovic, Volkhard Helms, Stephen High, Richard Zimmermann, Sven Lang

**Affiliations:** 1Medical Biochemistry and Molecular Biology, Saarland University, 66421 Homburg, Germany; 2School of Biological Sciences, Faculty of Biology, Medicine and Health, University of Manchester, Manchester M13 9PT, UK; 3Center for Bioinformatics, Saarland University, 66041 Saarbrücken, Germany; 4Bijvoet Center for Biomolecular Research, Utrecht University, 3584 CH Utrecht, The Netherlands

**Keywords:** differential protein abundance analysis, endoplasmic reticulum, guided entry of tail-anchored proteins, membrane proteins, protein targeting, protein translocation, Sec61 complex, signal recognition particle, SRP-independent targeting

## Abstract

Importing proteins into the endoplasmic reticulum (ER) is essential for about 30% of the human proteome. It involves the targeting of precursor proteins to the ER and their insertion into or translocation across the ER membrane. Furthermore, it relies on signals in the precursor polypeptides and components, which read the signals and facilitate their targeting to a protein-conducting channel in the ER membrane, the Sec61 complex. Compared to the SRP- and TRC-dependent pathways, little is known about the SRP-independent/SND pathway. Our aim was to identify additional components and characterize the client spectrum of the human SND pathway. The established strategy of combining the depletion of the central hSnd2 component from HeLa cells with proteomic and differential protein abundance analysis was used. The SRP and TRC targeting pathways were analyzed in comparison. TMEM109 was characterized as hSnd3. Unlike SRP but similar to TRC, the SND clients are predominantly membrane proteins with N-terminal, central, or C-terminal targeting signals.

## 1. Introduction

Importing proteins into the endoplasmic reticulum (ER) is an essential process for about 30% of the proteome of all nucleated human cells. This process involves two stages: the targeting of nascent or fully synthesized proteins to the ER and their subsequent insertion into or translocation across the ER membrane. Furthermore, the process relies on cleavable signal peptides (SPs) or non-cleavable targeting signals (i.e., transmembrane helices/TMHs) in the precursor polypeptides and dedicated components that read these signals and facilitate either ER targeting or membrane insertion and/or translocation. To date, four protein targeting pathways are known that deliver precursor polypeptides to the different ER membrane translocation complexes of human cells, specifically the Sec61 protein-conducting channel, as well as the membrane protein insertases EMC, Wrb/Caml, or the Sec61/TMCO1 supercomplex [1,2,3,4].

The first concept for protein targeting of the ER was established for the signal recognition particle/SRP and its receptor/SR in the ER membrane [5,6,7,8]. Accordingly, an N-terminal SP or TMH in the nascent precursor polypeptide is recognized by SRP in the cytosol, where it mediates a translational attenuation. The corresponding ribosome-nascent chain/RNC-SRP complex associates with the ER membrane via the heterodimeric SRP-receptor/SR [9,10,11]. Interaction between SRP and SR drives mutual GTP hydrolysis and triggers the release of the RNC at the Sec61 complex, Sec61/TMCO1, or EMC. Thus, in addition to its role in ER targeting of precursor polypeptides, SRP is a molecular chaperone for nascent precursor polypeptides and an mRNA-targeting device. 

The identification of precursor proteins with the ability for ER targeting independent of SRP—such as GPI-anchored membrane proteins in yeast, tail anchored (TA) membrane proteins in yeast and mammalian cells, and small presecretory proteins in the mammalian system—suggested alternative ER targeting machineries [12,13,14]. Bioinformatic analysis of the yeast secretome predicted up to 30% of all extracellular proteins as being SRP-independent. Subsequent studies in mammalian and yeast cells determined the capacity of the ER handling a broad variety of precursor proteins [15,16,17]. Their diversity is not restricted to differences in the primary structure of mature domains, but is equally evident in the amino acid sequence, length, hydrophobicity and location of the targeting signals [18]. Although each of those features impacts the targeting process, it is the location of the targeting signal within the precursor protein that led to the identification of the TRC-dependent targeting pathway for TA membrane proteins [1,13]. Based on the presence/absence of a cleavable SP and the location of a TMH within the primary structure, membrane proteins of the secretory pathway can be classified into four categories based on their first TMH. Type I membrane proteins have a cleavable SP preceding the TMH that orients in the N_out_–C_in_ orientation. Type II and type III membrane proteins lack a SP, and their first TMH inserts in the N_in_–C_out_ or N_out_–C_in_ orientation, respectively. TA proteins represent type IV membrane proteins and are described next.

TA proteins are single-spanning membrane proteins lacking a cleavable SP. Their TMH is located at the extreme C-terminus. This tail anchor inserts in a N_in_–C_out_ orientation so that the N-terminal domain of the protein faces the cytosol [1,13,19]. TA proteins of the secretory pathway, such as the β- and γ-subunits of the Sec61 complex, Cytochrome β_5_/Cytb5 and many components of vesicular transport, are targeted and inserted into the ER membrane. As for SRP-mediated targeting, TA proteins are chaperoned through the cytosol by cytosolic machinery and directed to the ER membrane via its heterodimeric ER membrane resident receptor. The minimal targeting machinery for TA proteins was termed as the TA receptor complex (TRC) in mammalian cells and the guided entry of tail-anchored proteins (GET) complex in yeast cells [15,16]. Briefly, the cytosolic ATPase TRC40, with its hydrophobic binding pocket, binds the TA protein, and the heterodimeric receptor complex, comprising Wrb and Caml, is required for efficient ER targeting [15,20,21,22]. The latter two proteins also facilitate the membrane insertion of the TA [23]. In addition, the mammalian TA-targeting machinery involves a ribosome-associating heterotrimeric Bag6 complex and SGTA, which act upstream of TRC40 [24,25].

Surprisingly, Get1, Get2, and Get3 yeast knockout strains are viable, indicating at least one additional ER targeting pathway. Indeed, a high-throughput screening approach identified a hitherto uncharacterized targeting pathway in yeast, termed the SRP-independent/SND system [17]. Three components were characterized and termed Snd1, Snd2, and Snd3. Two hallmarks of this targeting pathway were elucidated. First, similar to the SRP- and GET-targeting mechanisms, precursor substrates were targeted via a cytosolic mediator (Snd1) and its heterodimeric receptor in the ER membrane (Snd2, Snd3). Interestingly, Snd1 had previously been described as a ribosome-associated protein. Second, the SND pathway showed a preference for protein clients with a centrally located TMH. In addition, the SND pathway provides an alternative targeting pathway for substrates with a TMH at their N- or C-terminus, i.e., typical SRP- and TRC-dependent substrates. Sequence comparisons suggested the previously characterized ER membrane protein TMEM208 as a putative human Snd2 orthologue, which was termed hSnd2 [17,26]. Cell-free assays that combined siRNA-mediated *HSND2* gene silencing and protein import into the ER of semi-permeabilized HeLa cells demonstrated that hSnd2 has the same function as its yeast counterpart [27,28,29]. However, human orthologs of Snd1 and Snd3 have not been identified. Judging from the steady-state abundance of SR, Wrb/Caml, and hSnd2 in HeLa cells, the impression is that the SND pathway may account for more than 20% of precursor targeting in HeLa cells, whereas SR may account for around 75% and Wrb/Caml for just 2% [30]. 

Recently, ER targeting of so-called hairpin membrane proteins of lipid droplets (UBXD8) or of the ER membrane (RTN1, ARL6IP1) was found to involve cytosolic PEX19 and ER-membrane-localized PEX3 [31,32]. Therefore, an emerging concept for protein targeting to the ER is that a cytosolic molecular triage step occurs for ER-destined precursor polypeptides, and this event determines the fate of both nascent and fully synthesized polypeptides. This cytosolic triage process requires a complex network of targeting signals in nascent and completed polypeptide chains and a plethora of cytosolic factors that read these signals and direct them accordingly. However, these pathways are not fully separated from each other, i.e., some precursor polypeptides can be targeted by more than one pathway, at least in cell-free assays. Hence, some small human presecretory proteins (preproapelin) can be targeted to the Sec61 complex via the SRP-, SND-, and TRC-dependent pathways [29,33]. Likewise, some TA proteins (Cytb5, RAMP4, Sec61ß) can be targeted to the ER membrane via the same three pathways [27]. Thus, there is redundancy in these three targeting systems, i.e., the targeting pathways have overlapping substrate specificities and can partially substitute for each other. 

While there are fundamental insights available for the molecular mechanisms and client spectra of the SRP- and TRC-dependent pathways, our current knowledge of the SRP-independent/SND pathway is derived from yeast cells and mammalian cell-free assays. The aim of this study was to gain further insights into the components and client spectrum of the SND pathway in human cells under physiological conditions, including physiological translation rates and competing precursor polypeptides. For this, we used the established experimental strategy of combining the transient depletion of a central component of the pathway, here, hSnd2, with total cellular proteomic analysis by label-free quantitative mass spectrometry/MS and differential protein abundance analysis in HeLa cells [34,35,36,37,38]. For comparison, similar experiments were carried out for the SRP- and TRC-targeting pathways, for which different global approaches, such as ribosome profiling or component trapping data, are also available for human cells [39,40,41]. In addition, a combined hSnd2 and Wrb depletion was carried out to determine the expected client overlap between the two targeting pathways under physiological conditions. 

## 2. Materials and Methods

### 2.1. Materials

Rabbit antisera against TMEM109 and murine anti-ß actin antibodies, which served as a loading control for most Western blots, were purchased from Sigma-Aldrich (HPA011785, A5441) (St. Louis, MO, USA). Antibodies against REEP5 (14643-1-AP) were purchased from Proteintech. In addition, the following secondary antibodies were purchased: POD-coupled anti-rabbit antibodies from goat (Sigma-Aldrich 8275), POD-coupled anti-mouse antibodies from rabbit (Sigma-Aldrich A 9044), Cy5 conjugate of anti-rabbit IgG from goat (GE Healthcare PA45011), and Cy3 conjugate of anti-mouse IgG from goat (Sigma-Aldrich C 2181). Mouse monoclonal anti-OPG2 and goat polyclonal anti-LMNB1 (Santa Cruz sc-6217) antibodies, which served as a loading control for the amount of SP cells present per translation reaction, were described previously [42,43]. Additional rabbit antibodies were raised against the carboxy terminal (14-mer) or amino terminal IQ peptide (14-mer) of human Sec61α1, the amino terminal peptide of human Sec61β (9-mer), the carboxy terminal peptides of human SRα (10-mer), SRβ (12-mer), Sec62 (11-mer), TMEM109 (31-mer), the amino terminal peptides of human BiP (12-mer) and Grp170 (11-mer), the carboxy terminal peptides of human TRAPα (15-mer), Climp63 (12-mer), and hSnd2 (14-mer), and a mix of two peptides of human Caml, as described previously [28,44].

### 2.2. Cell Culture

HeLa cells (DSMZ no. ACC 57) were obtained from the German Collection of Microorganisms and Cell Cultures and cultivated at 37 °C in Dulbecco’s modified Eagle’s medium (DMEM; Gibco) containing 10% foetal bovine serum (FBS; Biochrom) and 1% penicillin/streptomycin (GE Healthcare) in a humidified environment with a 5% CO_2_ atmosphere. Cell growth and viability were routinely monitored using the Countess^®^ Automated Cell Counter (Invitrogen) according to the manufacturer’s instructions. The HeLa cells are replaced every five years and routinely tested for mycoplasma contamination by VenorGeM Mycoplasm Detection Kit (Biochrom AG, WVGM).

### 2.3. Depletion of Cells by siRNA Treatment

For most gene-silencing experiments, 5.2 × 10^5^ HeLa cells were seeded in a 6-cm culture plate in normal culture medium and then transfected with targeting siRNA (Table 1) or control siRNA (AllStars Negative Control siRNA, Qiagen, Hilden, Germany) to a final concentration of 20 nM using HiPerFect Reagent (Qiagen) as described previously [44,45]. After 24 h, the medium was changed, and the cells were transfected a second time. After an additional 48 h, silencing efficiencies were evaluated by Western blot analysis using the corresponding antibodies and a mouse anti-ß-actin antibody. The primary antibodies were visualized using goat anti-rabbit IgG-peroxidase conjugate and ECL^TM^, ECL^TM^ Plex goat anti-rabbit IgG-Cy5 or ECL^TM^ Plex goat anti-mouse IgG-Cy3 conjugate, and the Fusion SL (peqlab) luminescence imaging system or the Typhoon-Trio imaging system in combination with Image Quant TL 7.0 software (GE Healthcare).

### 2.4. Label-Free Quantitative Proteomic Analysis

Label-free quantitative proteomic analysis was carried out as previously described [34,35]. Briefly, HeLa cells (1 × 10^6^) were harvested, washed twice in PBS, and lysed in buffer containing 6M GnHCl, 20 mM tris(2-carboxyethyl)phosphine (TCEP; Pierce^TM^, Thermo Fisher Scientific, Waltham, MA, USA), 40 mM 2-chloroacetamide (CAA; Sigma-Aldrich) in 100 mM Tris, pH 8.0. The lysate was heated to 95 °C for 2 min and then sonicated in a Bioruptor sonicator (Diagenode) at the maximum power setting for 10 cycles of 30 s each. The entire process of heating and sonication was repeated once, and then the sample was diluted 10-fold with digestion buffer (25 mM Tris, pH 8, with 10% acetonitrile). Protein extracts were digested for 4 h with endoproteinase lysC, followed by the addition of trypsin for overnight digestion. After digestion, peptides were purified and loaded for mass spectrometric analysis. Raw data were processed using the MaxQuant computational platform [46]. The peak list was searched against Human Uniprot databases, with an initial precursor and fragment tolerance of 4.5 ppm. The match between the run feature was enabled, and proteins were quantified across samples using the label-free quantification algorithm in MaxQuant as label-free quantification (LFQ) intensities [47]. 

### 2.5. Co-Immunoprecipitation 

2 × 10^6^ HeLa cells were harvested and resuspended in 100 µL lysis buffer (20 mM HEPES/KOH pH 7.5, 400 mM KCl, 2 mM MgCl_2_, 1 mM EDTA, 2 mM DTT, 30 % (*v*/*v*) glycerine, 0.1 mM PMSF, 0.65% (*w*/*v*) CHAPS, 0.05 % (*v*/*v*) proteinase inhibitor cocktail PLAC. PLAC is a mixture of pepstatin A, leupeptin, antipain, and chymostatin (each 3 mg/mL) dissolved in DMSO). Cell lysis was carried out by light shaking for 45 min at 4 °C. Cell debris was separated from solubilized components by centrifugation at 190,000× *g* for 25 min at 4 °C (Optima™ MAX-E Ultrazentrifuge). A sample of the supernatant was kept for analysis (-IP). A total of 100 µL of the remaining supernatant was incubated over night at 4 °C with 2 µL of antibody serum (α-hSnd2 or α-TMEM109). The sample was incubated for 4 h at 4 °C with 5 % (*v*/*v*) of Protein A/G Sepharose (Pharmacia). Immunoprecipitation was performed in a cooled table top centrifuge for 2 min at 20,000× *g*. The supernatant was removed and a sample kept for analysis (+IP). Subsequently, the precipitated beads were washed three times with 200 µL lysis buffer (wash). Elution was performed by three iterative steps with 100 µL of elution buffer (200 mM glycine pH 2.2 in lysis buffer) added to each bead to release bound material. Each fraction was neutralized by adding 100 µL of 1 M Tris pH 7. The three eluate fractions were pooled and concentrated using TCA precipitation. Laemmli sampe buffer was added to the remaining beads and heated to release the remaining proteins. Similarly, samples of each fraction were mixed with Laemmli sample buffer and used for SDS-PAGE and Western blotting.

### 2.6. Creation of Plasmids for the Live Cell Protein–Protein Interaction

Primers were used to amplify the cDNA of interest by PCR and to add the specific cutting sites for restriction enzymes at the 5′ and 3′ ends. Primers used for PCR are listed in Table 2.

All reporter constructs were cloned in one of the four different backbone plasmids (X-N_S_, X-N_L_, X-C_S_, X-C_L_). These backbone plasmids as well as the negative control Halo-C_S_ are part of the NanoBiT PPI Starter System (Promega, Madison, WI, USA). To insert cDNA encoding for a protein of interest, a standardized workflow was used. For the insertion of cDNA that encodes for a protein of interest, the forward and reverse primers were designed that anneal to the 5′ and 3′ end of a cDNA (Table 2). Either the primer overhangs or the amplified PCR product included the required restriction sites for the multiple cloning site of the backbone vectors. PCR products were purified with the QIAquick PCR purification kit (Qiagen). PCR products and backbone plasmids were double digested for directed, sticky end insertion according to the manufacturer’s recommendation, based on the choice of restriction enymes (Thermo Fisher). Fragments were separated on a 1% agarose gel and purified via QIAquick gel extraction kit (Qiagen). The insert and the plasmid backbone were ligated in a 3:1 ratio for 1 h at room temperature using the T4 ligase (Thermo Fisher). JM101 *E. coli* cells were transformed with the ligated plasmid using a heat-shock at 42 °C for 90 s. Cells were plated on a 100 µg/mL ampicillin/LB-agar plate, and after 16 h, single colonies were picked to be grown in a 2 mL liquid culture using 100 µg/mL ampicillin/TB medium. The cultures were harvested via centrifugation in a table-top centrifuge (3000 rpm, 5 min). Plasmid DNA was purified using the PureYield Plasmid Miniprep System (Promega). Before use, all plasmid sequences were confirmed by sequencing (LGC Genomics) and subsequently amplified in the DH5α *E. coli* strain using heat shock transformation (42 °C for 45 s). Plasmids from a 100 mL ampicillin/LB medium culture were purified using the Plasmid Midi Kit (Qiagen).

### 2.7. Live cell Protein–Protein Interaction (PPI) Using the NanoBiT Assay 

A total of 20.000 cells were seeded using 100 µL media per cavity of a white, flat bottom 96-well plate (GreinerBioOne) and cultivated for 24 h in the incubator at 37 °C and 5% CO_2_. The transfection with reporter constructs was performed without prior media exchange. The stock concentration of each plasmid was adjusted to 100 ng/µL. For the PPI experiment, 0.5 µL of the plasmids encoding the LgBiT and SmBiT fusion proteins were mixed with 6.72 µL Opti-MEM without phenol red (Thermo Fisher), and 0.28 µL FuGENE HD (Promega) was added. For the PPI competition experiment, 0.5 µL of the three plasmids encoding the LgBiT and SmBiT fusion proteins as well as one of the interaction partners without a tag were mixed with 6.08 µL Opti-MEM without phenol red, and 0.42 µL FuGENE HD (Promega) was added. All transfection mixes were incubated at room temperature for 10 min and added to the wells. Transfected cells were incubated for 24 h. Five min before the start of the luminescence reading, the medium was replaced with pre-warmed Opti-MEM without phenol red. The 96-well plate was placed in the preheated microplate reader (Tecan Infinite M200). The settings for recording luminescence were as follows: interval 1 min; shaking before every interval; integration time 1000 ms; settle time 50 ms. A total of 20 µL of the 1:20 diluted Nano-Glo Live Cell Assay System (Promega) was added according to the manufacturer’s protocol to initiate bioluminescence, which was recorded for 9 min. 

### 2.8. Cell-Free Protein Transport Experiments

All cDNAs encode genes of human origin modified with one or two N- or C- terminal OPG2 tag(s) (residues 1-18 or 1-26 of bovine rhodopsin (Uniprot: P02699)), unless specified otherwise. Cytb5OPG2, GypCOPG2, BCMAOPG2 (containing an artificial N-glycan site, Y13T), and rat Syt1OPG2 were all as previously described [42]. Linear DNA templates were generated by PCR, and mRNA was transcribed using T7 or SP6 polymerase. Primers (Integrated DNA Technologies) used for PCR are listed in Table 3.

HeLa cells (mycoplasma-free), as previously described [48], were provided by Martin Lowe (University of Manchester, UK) and were cultured in DMEM supplemented with 10% (*v*/*v*) FBS and maintained in a 5% CO_2_ humidified incubator at 37 °C. Knockdown of the target genes and preparation of SP cells was performed as previously described [42,43]. Briefly, 1 × 10^6^ cells were seeded per 10 cm^2^ dish and, 24 h after plating, the cells were transfected with either control siRNA (ON-TARGETplus Non-targeting control pool; Dharmacon), *SRPRA* siRNA (SRα-kd, GE Healthcare, sequence GAGCUUGAGUCGUGAAGACUU) [27,42], hSnd2 siRNA (hSnd2-kd, GE Healthcare, sequence CUAUAGGGUCGUUGAAUAATT) [27,28], TMEM109 siRNA (TMEM109-kd, GE Healthcare, sequence CAGGTTTGATGTGGAATCACA), or a combinations thereof at a final concentration of 20 nM using INTERFERin (Polyplus, 409-10) as described by the manufacturer. A total of 72 h post-initial transfection, cells were detached by incubation with 3 mL of 0.25% trypsin-EDTA solution (Sigma-Aldrich) for 10 min at RT, mixed with 4 mL of KHM buffer (110 mM KOAc, 2 mM Mg(OAc)_2_, 20 mM HEPES-KOH pH 7.2) supplemented with 100 µg/mL Soybean trypsin inhibitor (Sigma-Aldrich, T6522) and centrifuged at 250× *g* for 3 min at 4 °C. The pellet was resuspended in 4 mL KHM buffer supplemented with 80 µg/mL high purity digitonin (Calbiochem), and the cells were incubated on ice to selectively permeabilize the plasma membrane. After 10 min, the cells were again centrifuged at 250× *g* for 3 min before resuspension in HEPES buffer (90 mM HEPES, 50 mM KOAc, pH 7.2) and incubation on ice for 10 min. Cells were pelleted by centrifugation once more and resuspended in 100 µL KHM buffer, and the endogenous mRNA was removed by treatment with 0.2 U Nuclease S7 Micrococcal nuclease, from *Staphylococcus aureus* (Sigma-Aldrich, 10107921001), and 1 mM CaCl_2_ at RT for 12 min before quenching by the addition of EGTA for a 4 mM final concentration. Cells were centrifuged at 13,000× *g* for 1 min and resuspended in an appropriate volume of KHM buffer to give a suspension of 3 × 10^6^ SP cells/mL as determined by trypan blue staining (Sigma-Aldrich, T8154). SP cells were included in translation master mixes such that each translation reaction contained 2 × 10^5^ cells/mL.

All samples were denatured for 1–24 h at 37 °C prior to resolution by SDS-PAGE (10% or 16% PAGE, 120 V, 120 min). To analyze the translation products, gels were fixed for 5 min (20% (*v*/*v*) MeOH, 10% (*v*/*v*) AcOH) and dried for 2 h at 65 °C, and the radiolabeled species were visualized using a Typhoon FLA-700 (GE Healthcare) following exposure to a phosphorimaging plate for 24–72 h. Knockdown efficiencies (SRα, hSnd2, TMEM109) and controls (LMNB1, EMC2, EMC6, Sec61α, CAML, SRP54) were determined by quantitative immunoblotting. Following transfer to a PVDF membrane in transfer buffer (0.06 M Tris, 0.60 M glycine, 20% MeOH) at 300 mA for 2.5 h, PVDF membranes were incubated in 1X Casein blocking buffer (10X stock from Sigma-Aldrich, B6429) made up in TBS, incubated with appropriate primary antibodies (1:500 or 1:1000 dilution) and processed for fluorescence-based detection as described by LI-COR Biosciences using appropriate secondary antibodies (IRDye 680RD Donkey anti-Goat, IRDye 680RD Donkey anti-Rabbit, IRDye 800CW Donkey anti-Guinea pig, IRDye 800CW Donkey anti-Mouse) at 1:10,000 dilution. Signals were visualized using an Odyssey CLx Imaging System (LI-COR Biosciences).

Translation and membrane insertion assays (30 μL) were performed in nuclease-treated rabbit reticulocyte lysate (Promega) in the presence of EasyTag EXPRESS ^35^S Protein Labelling Mix containing [^35^S] methionine (Perkin Elmer) (0.533 MBq; 30.15 TBq/mmol), 25 μM amino acids minus methionine (Promega), 6.5% (*v*/*v*) siRNA-treated SP cells, and ~10% (*v*/*v*) of mRNA transcribed in vitro encoding the relevant OPG2-tagged precursor protein for 1 h at 30 °C. Following incubation with 0.1 mM puromycin for 10 min at 30 °C to ensure translation termination and ribosome release of newly synthesized proteins, the total reaction material was then diluted with nine volumes of Triton immunoprecipitation buffer (10 mM Tris-HCl, 140 mM NaCl, 1 mM EDTA, 1% (*v*/*v*) Triton X-100, pH 7.5) supplemented with 5 mM PMSF and 1 mM methionine, and the samples were incubated under constant agitation with antisera recognizing the OPG2-tag (1:200 dilution) for 16 h at 4 °C to recover both the membrane-associated and non-targeted nascent chains. Samples were then incubated under constant agitation with 10% (*v*/*v*) Protein-A-Sepharose beads (Genscript) for a further 2 h at 4 °C before recovery by centrifugation at 13,000× *g* for 1 min. Protein-A-Sepharose beads were washed with Triton immunoprecipitation buffer prior to suspension in SDS sample buffer.

### 2.9. Carbonate Extraction

After in vitro synthesis, 5 µL of the translation mix was centrifuged with 100,000× *g* for 5 min at 4 °C through a 100 µL sucrose cushion (500 mM sucrose, 50 mM HEPES/KOH pH 7.6, 150 mM KOAc, 2 mM MgOAc, 1 mM DTT) using the TLA100.3 rotor. While 30 µL of the supernatant (S1) were kept for ammonium sulfate precipitation, the pellet was resuspended in 30 µL alkaline carbonate extraction buffer (100 mM Na_2_CO_3_ pH 11) and kept on ice for 15 min. The resuspended pellet was centrifuged with 130,000× *g* for 10 min at 4 °C through a 100 µL alkaline sucrose cushion (250 mM sucrose, 100 mM Na_2_CO_3_ pH 11) using the TLA100.3 rotor. A total of 30 µL of the supernatant (S2) was kept for alkaline carbonate precipitation. The pellet (P) as well as the precipitates of S1 and S2 were resuspended in 20 µL Laemmli sample buffer and heated for 15 min at 65 °C. In addition, an additional 5 µL of the translation mix (T) were mixed with 15 µL Laemmli sample buffer and heated for 15 min at 65 °C. All samples were separated by SDS-PAGE. Gels were dried and the radiolabeled products visualized using Typhoon-Trio imaging system (GE Healthcare) following exposure to a phosphorimaging plate for 24–72 h.

### 2.10. Peptide Spot Assay

Peptides, corresponding to the complete sequence of hSnd2 or TMEM109 with a length of 15 amino acid residues each, were synthesized by a ResPep SL (Intavis) fully automated peptide synthesizer on a derivatized cellulose membrane via their C-terminal ends. The amino acid frame was shifted by 3 amino acids from one spot to the following spot [49]. Membranes were equilibrated in binding buffer (1% BSA in 150 mM NaCl, 50 mM Tris/HCl pH 7.5, 0.1% Triton X-100) for 1 h at RT to minimize unspecific binding. The digitonin extract from canine pancreatic rough microsomes [50] was diluted into binding buffer (1:3) and incubated at 4 °C overnight with the membranes. The membranes were washed excessively with binding buffer for 10 min each and, thereafter, incubated with antibodies against TMEM109 and hSnd2, respectively. For detection of the bound primary antibodies, POD-coupled secondary anti-rabbit antibodies (from goat) were used, and the membrane was subjected to luminescence imaging using ECL^TM^ (GE Healthcare) and a Fusion SL imaging device (PEQLAB).

### 2.11. Statistical Analysis of MS Experiments

Data analysis was carried out as previously described [34,35]. Briefly, each of the MS experiments provided proteome-wide abundance data as LFQ intensities for three sample groups—one control (non-targeting siRNA-treated) and two stimuli (down-regulation by two different targeting siRNAs directed against the same gene)—each having three data points (replicates). Only proteins that were detected in both experiments (repeats) were considered. Missing data points were generated by imputation as previously described [34,35]. To identify which proteins were affected by knockdown in siRNA-treated cells relative to the non-targeting (control) siRNA-treated sample, we log2-transformed the ratio between siRNA and control siRNA samples, and performed two separate unpaired *t*-tests for each siRNA against the control siRNA sample. The *p*-values obtained by unpaired *t*-tests were corrected for multiple testing using a permutation false discovery rate (FDR) test. Proteins with an FDR-adjusted *p*-value of below 5% were considered significantly affected by the knockdown of the targeted proteins. The results from the two unpaired *t*-tests were then intersected for further analysis, meaning that the abundance of all reported candidates was statistically significantly affected in both siRNA silencing experiments in the same direction (up or down). All statistical analyses were performed using the R package SAM (https://library.stanford.edu/projects/r/stanford-r-packages; accessed 12 September 2022) [51].

Protein annotations of signal peptides, transmembrane regions, and N-glycosylation sites in humans and yeast were extracted from UniProtKB entries using custom scripts [34,35]. The enrichment of functional Gene Ontology annotations (cellular components and biological processes) among the secondarily affected proteins was computed using the GOrilla package [52]. Using custom scripts, we computed the hydrophobicity score and glycine/proline (GP) content of SP and TMH sequences [34,35]. A peptide’s hydrophobicity score was assigned as the average hydrophobicity of its amino acids according to the Kyte–Doolittle propensity scale (averaged over the sequence length) [53]. GP content was calculated as the total fraction of glycine and proline in the respective sequence. ΔG_app_ values of SP and TMH were calculated with the ΔG_app_ predictor for TM helix insertion (https://dgpred.cbr.su.se/index.php?p=home; accessed 12 September 2022).

SP segmentation prediction was carried out as previously described [34,35], using the well-established prediction tool Phobius [54] to identify the N-region, H-region, and C-region of all SP. Based on this, we calculated the total net charge of the N-region, the polarity of the C-region, and the hydrophobicity and absolute length of the H-region. The polarity score of a single peptide was calculated as the averaged polarity of its amino acids according to the polarity propensity scale derived by Zimmerman et al. [55]. The hydrophobicity score was calculated in the same fashion using the Kyte–Doolittle propensity scale [53].

The full dataset of human mRNA coding sequences (CDS) was downloaded from https://www.ensemble.org/index.html; accessed 31 March 2018. It contains 110,788 CDS records. To avoid a bias in the comparison between the CDS of candidates against the full human CDS (background dataset), we only used the CDS transcripts for the proteins identified in the MS data. For this, we translated the full CDS dataset into protein sequences and compared the resulting proteins with the MS data. A total of 4660 (out of 4856) proteins could be matched in this way. The goal of this analysis is to compare the (estimated) translation rates of the peptide sequences negatively affected by knockdown which contain either a signal peptide or an N-terminal TMH to the estimated translation rates of the proteins in the full MS dataset. As we are only interested in the N-terminal part of the sequence, and to ensure comparability of the results independent of the length of the actual mRNA sequences, we always used the first 240 nucleotides (which is equivalent to 80 amino acids) of the CDS for the subsequent analysis. The translation rates were computed based on the codon-specific elongation rates provided by Trösenmeier et al. [56]. The rates determined there describe the specific elongation speed for each codon during the translation process. Here, the rates were inverted and summed up, resulting in the (estimated) translation speed for the N-terminal 240 nt of each sequence (240 nt.~80 codons). The two distributions were compared using a nonparametric Wilcoxon test. 

### 2.12. Graphical Representation and Statistical Analysis of Cell-Free Protein Transport

Bar graphs depict either the efficiencies of siRNA-mediated knockdown in SP cells calculated as a proportion of the protein content when compared to the untargeted control siRNA or membrane insertion efficiencies calculated as the ratio of N-glycosylated protein to the amount of non-glycosylated protein, with all control samples set to 100%. All replicates were performed using individual siRNA-mediated knockdowns (n = 3, biologically independent experiments). Normalized values were used for statistical comparison (one-way ANOVA, Dunnett’s multiple comparisons test). Statistical significance is given as *p* < 0.05 (*); *p* < 0.01 (**); *p* < 0.001 (***).

## 3. Results

### 3.1. Quantitative Proteomic Analysis of HeLa Cells after Depletion of SRα by siRNA Identifies Precursors with N-Terminal Targeting Signals as Predominant SRα Clients 

As a proof of concept for investigating precursor polypeptide targeting to the human ER, we applied an established experimental strategy to identify precursor polypeptides that depend on SR and, therefore, SRP in their ER targeting [34]. First, HeLa cells were treated in triplicates with two different *SRA*-targeting siRNAs (one of which was previously established) in parallel to a non-targeting (control) siRNA for 96 h in two independent experiments. Then, label-free quantitative MS and, subsequently, differential protein abundance analysis, provided proteome-wide abundance data as LFQ intensities for three sample groups for each experiment—one control (non-targeting siRNA-treated) and two stimuli (down-regulation by two different targeting siRNAs directed against the same gene)—each having six data points (Figure 1A). To identify which proteins were affected by knockdown in siRNA-treated cells relative to the non-targeting control, we log2-transformed the ratio between siRNA and control siRNA samples and performed two separate unpaired *t*-tests for each siRNA against the control siRNA sample [34]. The *p*-values obtained by unpaired *t*-tests were corrected for multiple testing using a permutation false discovery rate (FDR) test. Proteins with an FDR-adjusted *p*-value below 5% were considered significantly affected by the knockdown of the targeted protein. After SRα depletion, 7044 different proteins were quantitatively detected by MS in the two experiments, 4856 of which were detected in all samples (Figure 1A, Appendix A). The MS data have been deposited to the ProteomeXchange Consortium (http://www.proteomexchange.org; accessed on 12 September 2022). They included the expected representation of proteins of the endocytic and exocytic pathways (29%) and proteins with cleaved SP (7%), N-glycosylated proteins (10%), and membrane proteins (14%) (Figure 1A, left pies), which were comparable to a previous Sec61α depletion experiment [34]. Applying the established statistical analysis, we found that transient and partial SRα depletion significantly affected the steady-state levels of 139 proteins: 133 negatively and 6 positively (permutation-false-discovery-rate-adjusted *p*-value < 0.05). Among the negatively affected proteins, Gene Ontology (GO) terms assigned 50% to organelles of the endocytic and exocytic pathways, representing a strong enrichment compared to the value for the total quantified proteome (26%) (Figure 1A, large right pie). We also detected a significant enrichment of precursor proteins with SP (2.5-fold), N-glycosylated proteins (2.8-fold), and membrane proteins (2.5-fold) (Figure 1A, small right pies). As expected, SRα (encoded by the SRPRA gene) itself was negatively affected (Figure 1A, volcano plots), which was confirmed by Western blot (Appendix A). As was SRβ negatively affected, consistent with the previous observation that the subunits of ER membrane protein complexes such as Sec61 and TRAP are degraded in the absence of the Sec61α and TRAPβ, respectively [34]. No ubiquitin-conjugating enzymes or potential compensatory ER protein-import components were detected among the positively affected proteins (Appendix A), consistent with the relatively low number of negatively affected proteins.

The negatively affected proteins included 24 proteins with a cleavable SP, among them 14 single-spanning type I membrane proteins (Figure 2A,B, Appendix A). In addition, there were 30 membrane proteins without an SP among the negatively affected proteins, including 3 ER hairpin membrane proteins (ATL2, CAV1, and REEP3), 10 single-spanning membrane proteins with apparent delta G values for membrane insertion between −5.685 and 3.594 kcal/mol (such as the peroxisomal protein PEX3), and 17 multi-spanning membrane proteins (Figure 2C, Appendix A). Of these 54 negatively affected proteins, 33 are N-glycosylated (including 20 proteins with SP). Thus, the precursors of the negatively affected proteins with SP and TMH can be expected to be clients of the SRP/SR targeting pathway. As expected, there were no TA proteins found among the SRα clients. The fact that 44% of these SR clients have an N-terminal SP and 77% of the remaining membrane protein clients have comparatively N-terminal TMHs (Figure 2G and Appendix A) is entirely consistent with previous proximity-based ribosome profiling studies [39,40]. 

### 3.2. Quantitative Proteomic Analysis of HeLa Cells after Depletion of Wrb by siRNA Identifies Various Types of Precursors as Wrb Clients

As a second proof of principle, and for direct comparison with the two following experiments, we applied the same established experimental strategy to identify precursor polypeptides that depend on Wrb in their targeting to the human ER [34]. HeLa cells were treated in triplicate with two different *WRB*-targeting siRNAs (one of which was previously established) in parallel to a non-targeting (control) siRNA for 96 h. Applying the same MS plus analysis protocol as described above, after Wrb depletion, 6,052 different proteins were quantitatively detected by MS in all samples (Figure 1B, Appendix A). The MS data have been deposited to the ProteomeXchange Consortium. The dataset of detected proteins showed the expected representation of proteins with cleaved SP, N-glycosylated proteins, and membrane proteins (Figure 1B, left small pies) [34]. Applying the established statistical analysis, we found that transient and partial Wrb depletion significantly affected the steady-state levels of 296 proteins: 144 negatively and 152 positively (permutation-false-discovery-rate-adjusted *p*-value < 0.05). Among the negatively affected proteins, GO terms assigned 45% to organelles of the endocytic and exocytic pathways, representing a strong enrichment compared to the value for the total quantified proteome (28%) (Figure 1B, large right pie). We also detected a small enrichment of precursor proteins with SP (1.4-fold) and membrane proteins (1.4-fold) (Figure 1B, small right pies). Wrb was not quantified by MS, but its depletion was confirmed by Western blot (Appendix A). Furthermore, Wrb depletion was also indicated by the depletion of its interaction partner Caml, encoded by the CAMLG gene (Figure 1B, volcano plots). Notably, two cytosolic Wrb interaction partners were also negatively affected (ASNA1/GET3/TRC40, GET4/TRC35). Among the positively affected proteins, there were three ubiquitin-conjugating enzymes (ARIH1, MID1, RNF25), four cytosolic molecular chaperones (DNAJB2, DNAJC12, HSBP1, HSPB8) and, in the case of one of the two *WRB*-targeting siRNAs, the two subunits of the SRP receptor (Figure 1B, Appendix A).

The negatively affected proteins, i.e., putative Wrb and TRC clients, included 13 proteins with cleavable SP (including 6 type I membrane proteins) and 14 membrane proteins without SP (Figure 2A, Appendix A). Among the latter were 3 TA membrane proteins (GOLGA5, UBE2J2, and the peroxisomal FAR1), 2 single-spanning membrane proteins, 8 multi-spanning membrane proteins, and the ER hairpin protein REEP3 (Figure 2A, Appendix A). Of these 27 negatively affected proteins, 13 are N-glycosylated (including 11 proteins with SP), which is consistent with the reports that pathogenic variants of Caml as well as TRC35 or TRC40 are linked to congenital disorders of glycosylation in human patients [57]. Thus, similar to SRP/SR, 48% of the observed Wrb clients have N-terminal SP, and 57% (i.e., 8 out of a total of 14) of the remaining membrane protein clients have more C-terminal TMHs (Appendix A). This is partially consistent with previous results from cellular TRC trapping and in vivo experiments, which suggested that there is a preference of TRC for C-terminal targeting signals, typically found in TA membrane proteins such as GOLGA5 [41].

### 3.3. Quantitative Proteomic Analysis of HeLa Cells after Depletion of hSnd2 by siRNA Identifies Precursors of Membrane Proteins as Predominant hSnd2 Clients

Next, we applied the same established experimental strategy to identify precursor polypeptides that depend on hSnd2 for their targeting to the ER in two different experiments [34]. In the first experiment, HeLa cells were treated in triplicate with two different previously established *HSND2*-targeting siRNAs in parallel to a non-targeting (control) siRNA for 96 h. According to the same MS plus abundance analysis protocol as described above, 5997 different proteins were quantitatively detected by MS in all samples after hSnd2 depletion (Figure 3A, Appendix A). The MS data have been deposited to the ProteomeXchange Consortium. They included the expected representation of proteins of the endocytotic and exocytotic pathways and proteins with cleaved SP, N-glycosylated proteins, and membrane proteins (Figure 3A, left pies) [34]. Applying the established statistical analysis, we found that transient and partial hSnd2 depletion significantly affected the steady-state levels of 76 proteins: 43 negatively and 33 positively (permutation false discovery rate-adjusted *p*-value < 0.05). Among the negatively affected proteins, GO terms assigned roughly 47% to organelles of the endocytic and exocytic pathways, representing a strong enrichment compared to the value for the total quantified proteome (27%) (Figure 3A, large right pie). In contrast to the enrichment of proteins with cleavable SPs after *SRA* or *WRB* silencing, no enrichment of SP-carrying clients was observed after the depletion of hSnd2. However, we did detect a small enrichment of N-glycosylated proteins (1.4-fold) and a more pronounced one for membrane proteins (2.4-fold) (Figure 3A, small right pies). hSnd2 itself was not quantified by MS (Figure 3A, volcano plots), but its depletion was confirmed by Western blot (Appendix A). In line with the relatively mild negative effects, there was only one ubiquitin-conjugating enzyme, two ubiquitin-related proteins, and no compensatory ER protein-import components among the positively affected proteins (Appendix A).

The negatively affected proteins, i.e., potential hSnd2 clients, included 3 proteins with cleavable SP (all being single-spanning type I membrane proteins) and 9 membrane proteins without SP, including 2 TA membrane proteins (VAMP4, VAMP8), 2 single-spanning membrane proteins with apparent delta G values for membrane insertion ranging from 1.571 to −3.435 kcal/mol (MYO9A, PTGIS), and 5 multi-spanning membrane proteins (such as TRPM7), thus confirming previously observed classes of hSnd2 clients (TRPC6, Cytb5) (Figure 2A, Appendix A). Of these 12 negatively affected proteins, 4 are N-glycosylated (including 1 protein with SP). Strikingly, all hSnd2 clients which were identified under these cellular conditions are membrane proteins and, unlike SRP/SR clients, only 25% of these hSnd2 clients have an N-terminal SP. In further contrast to typical SRP/SR-dependent precursors, 44% of the remaining membrane protein clients have more C-terminally located TMHs (Appendix A), thereby displaying some similarity to the TRC pathway clients as previously observed for the yeast SND-targeting pathway [17].

### 3.4. Quantitative Proteomic Analysis of HeLa Cells after Simultaneous Depletion of hSnd2 and Wrb Identifies Precursors with More C-Terminal Targeting Signals as Predominant Clients of Both Components

Based on the observed overlap in clients between the SND- and TRC-dependent pathways, both in yeast and mammalian cell-free systems, we applied the established experimental strategy to directly identify precursor polypeptides that may involve hSnd2 as well as Wrb in targeting to the human ER [34]. The experiment was carried out under conditions of simultaneous depletion in parallel to the above described single hSnd2 depletion. Applying the same MS plus abundance analysis protocol as described above, after hSnd2 plus Wrb knockdown, 5997 different proteins were quantitatively detected by MS in all samples (Figure 3B, Appendix A). The MS data have been deposited into the ProteomeXchange Consortium. Applying the established statistical analysis, we found that simultaneous hSnd2 and Wrb depletion significantly affected the steady-state levels of 401 proteins: 196 negatively and 205 positively (permutation false discovery rate-adjusted *p*-value < 0.05). Among the negatively affected proteins, GO terms assigned roughly 38% to organelles of the endocytic and exocytic pathways, representing a small enrichment compared to the value for the total quantified proteome (27%) (Figure 3B, large right pie). We also detected a small enrichment of N-glycosylated proteins (1.6-fold) and a larger one of membrane proteins (2.0-fold) (Figure 3B, small right pies). As was to be expected, Caml, the partner subunit of Wrb, was negatively affected (Figure 3B, volcano plots), as were hSnd2 and Wrb according to Western blot (Appendix A). Among the positively affected proteins, there were ubiquitin-conjugating enzymes (RING1, UBE2G2) and ubiquitin-related proteins (NEDD8, SUMO2, UBE2F, UBE2I, ZNF451), a cytosolic molecular chaperone (HSPA12A), a mitochondrial protein import component (TOMM20), and the ER protein import component α-subunit of the TRAP complex, all consistent with an accumulation of precursor polypeptides in the simultaneous absence of hSnd2 and Wrb (Appendix A).

For further analysis of the negatively affected proteins, only those clients which had not been observed in one of the single depletion experiments were considered as clients of both pathways (shown in the respective overlaps in Figure 2A, Venn diagram). The putative hSnd2 plus Wrb clients included 13 proteins with cleavable SP, including 5 type I membrane proteins (DSG2, HLA-A, HLA-B, IL6ST, SDC4) and the 2 multi-spanning membrane proteins SLC39A6 and TMEM109, plus the 6 soluble proteins DCD, GANAB, LGMN, LRPAP1, PRSS21, NS SMOC1, and 30 membrane proteins without SP (not counting Caml) (Figure 2A, Appendix A). Among the latter were 5 TA membrane proteins (C4orf3, EMD, JPH1, STX2, STX3), the hairpin protein REEP5, 3 single-spanning membrane proteins with apparent delta G values for membrane insertion ranging from 2.128 to −1.979 kcal/mol (MXRA7, PLD3, POMK), and 21 multi-spanning membrane proteins, including SEC62 and STX17 (Appendix A). Of these 43 negatively affected proteins, 22 are N-glycosylated (including 11 proteins with SP). In contrast to SRP/SR (44%), 30% of these hSnd2 plus Wrb clients have an N-terminal SP. In additional contrast to SRP/SR, 40% of the remaining membrane protein clients have more C-terminal TMHs (Appendix A), which is also consistent with results for the yeast SND targeting pathway [17]. Furthermore, these double knockdown data support previous results from cellular TRC trapping experiments, which characterized the proteins EMD and Syntaxins 5 and 6 as clients of the TRC/WRB pathway [41], and considerably extended the client spectrum of the TRC pathway, as will be discussed next. 

### 3.5. Characterization of Putative hSnd2 and Wrb Clients

When the targeting signals (SPs and TMHs) of clients of the hSnd2 and the other two targeting pathways were analyzed with respect to hydrophobicity, delta G for membrane insertion, helix propensity, or after the segmentation of SPs into THE N-, H-, and C-region according to established procedures [33,34,35], no significantly distinct features became apparent (Appendix A). Furthermore, we did not detect any striking differences in translation speeds between the various clients when the first 80 codons were analyzed according to Trösenmeier et al. [56] (Appendix A).

For further characterization of the putative clients of the hSnd2 and Wrb pathways, clients identified in the respective single depletion experiments were pooled with those from the double depletion experiment. This brought the total numbers of clients to 55 and 70, respectively, which is similar to the SRα depletion experiment (n = 54) and, therefore, suitable for a more detailed analysis (Appendix A, sum). Under these conditions, comparatively few hSnd2 clients have an N-terminal SP (29%), as compared to SRα (44%) and Wrb clients (37%) (Figure 2B). In additional contrast to the SRα, but comparable to the TRC pathway, 42% of the remaining membrane protein clients have more C-terminal TMHs (compared to 24% in the case of SRα and 47% in the case of Wrb) (Figure 2G and Appendix A). Furthermore, the preference of the human SND pathway for membrane proteins that was observed in the single depletion experiment was confirmed, with almost 90% of the hSnd2 clients identified being membrane proteins (Figure 2A,B, Appendix A). SRα and Wrb clients do not show the same pronounced enrichment of membrane proteins (about 80%). When the hSnd2 and Wrb clients with TMHs were analyzed in more detail, the multi-spanning membrane proteins stood out as predominant clients in both cases, being 47% and 41% of total clients, respectively, as compared to only 32% in the case of SRα (Figure 2C versus Figure 2D,E, Appendix A). Thus, both pathways appear to share a preference for multi-spanning membrane proteins. This is particularly remarkable for the TRC pathway, which, to date, has primarily been associated with the ER targeting of TA proteins [1], notwithstanding some first indications of a link with multi-spanning membrane protein biogenesis in the mammalian system [58] and a link with hairpin protein biogenesis in yeast [59]. Furthermore, more hSnd2 as well as Wrb membrane protein clients have relatively central TMHs as compared to SRα-dependent membrane proteins. This is readily apparent when the data are presented as violin plots (Figure 2G) and further supported by the fact that the average relative position of the first TMH within the amino acid sequence lies between 32.8+/−5.2% and 35+/−6% for hSnd2 and Wrb clients, but at 21+/−4.3% for SRα membrane protein clients. This distribution is partially due to the fact that there are TA proteins among the hSnd2 and Wrb clients, which is not the case for SRα clients (Figure 2G and Appendix A). Moreover, this is consistent with the observation that there is little overlap between the SRα clients and clients of the other two pathways, i.e., major histocompatibility complex proteins (HLA), ITPR3, REEP3, TMEM41B, and TMX3 (Figure 2A).

### 3.6. Validation of Putative Clients Confirms Conclusions from MS Experiments

To validate the proteomic data regarding putative hSnd2 substrates and possible compensatory components, we conducted independent Western blot experiments with targeting and control siRNA-treated HeLa cells. *WRB* plus *HSND2* silenced cells were generated as described for the MS experiments above. The three proteins REEP5, Sec62, and TMEM109 were analyzed as potential clients (Figure 4). Notably, their selection was purely based on the availability of reliable antibodies. The representative blots fully confirmed the proteomic analysis and verified REEP5 as well as Sec62 as clients of both targeting pathways. The depletion of TMEM109 after simultaneous Wrb and hSnd2 depletion was also confirmed and will be discussed next. Furthermore, the blots confirmed the potential compensatory overproduction of TRAPα after simultaneous Wrb and hSnd2 depletion and the overproduction of SRα and SRβ, which had only been detected as significant after Wrb depletion in the MS experiment (Figure 4 and Appendix A, Appendix A).

### 3.7. The ER Membrane Protein TMEM109 Interacts with the Targeting Receptor hSnd2 and Components of the Sec61 Translocon 

As stated above, orphan subunits of heteromultimeric protein complexes, including the Wrb/Caml or SRα/SRβ targeting receptors (Figure 1), are degraded in the absence of one of the main components [28,60,61,62,63]. The same phenomenon was also previously confirmed by our MS approach for the Sec61 and TRAP complex [34]. We, therefore, took advantage of the proteomic datasets presented here and searched for additional candidate components of the mammalian SND pathway. Specifically, we searched for proteins that showed a significant (*p* < 0.001) and more than 50% reduced abundance (i) upon hSnd2 depletion with both siRNAs, (ii) upon hSnd2 plus Wrb double depletion with both siRNA combinations, and (iii) for proteins that were identified as potential hSnd2 interaction partners in the co-immunoprecipitation dataset previously published by Haßdenteufel et al. [28]. Considering those parameters, the ER membrane protein TMEM109 was found as the single candidate (Figure 5A, Appendix A). To verify a physical protein–protein interaction between hSnd2 and TMEM109, three approaches were used: reciprocal co-immunoprecipitation, immobilized peptide array interaction, and bimolecular luminescence complementation in living cells (NanoBiT). 

To avoid false-positive results from plasmid-driven overexpression or the addition of protein tags, specific antibodies recognizing cellular hSnd2 and TMEM109 were raised and used for the co-immunoprecipitation and Western blot detection of both membrane proteins. TMEM109 and hSnd2 could be solubilized and immunoprecipitated from lysates of untreated HeLa cells, and analysis of the eluted fractions showed that TMEM109 co-immunoprecipitated with hSnd2 and vice versa (Figure 5B and Appendix A). Grp170 and Climp63 were used as negative control proteins for the ER lumen and ER membrane, respectively, and neither one co-eluted with hSnd2 or TMEM109. In silico structural prediction using Alphafold and other sequence-based algorithms that evaluate protein topology of a protein suggest that hSnd2 contains four TMHs [54,64,65], with its N- and C-termini located in the cytosol (Figure 5C). TMEM109 has a cleavable SP and was shown experimentally to contain three TMHs [53]. However, sequence-based algorithms including TOPCONS and Phobius predict four TMHs, whilst the Alphafold structural model is challenging to align into a membrane (Figure 5C), making this scarcely characterized membrane protein an interesting candidate for further structural analyses [66,67]. In silico analysis of the TMEM109 primary structure identified a putative coiled-coil domain in the cytosolic C-terminus, which is reminiscent of Wrb, EMC3, and TMCO1 [68,69]. Interestingly, we also identified a short oligopeptide (_224_EELRWRQRR_232_) located within this coiled-coil domain that shows sequence similarity to an oligopeptide in the cytosolic C-terminal domain of hSnd2 (_156_EHNEKRQRR_164_) (Figure 5C).

Next, peptide spot arrays that display the full primary structure of TMEM109 or hSnd2 as 15 separate amino acid fragments were used to identify the regions of interaction between both proteins (Figure 5D). The peptide spot membranes were incubated with digitonin-solubilized rough microsomes (RM), and after washing, the binding of solubilized TMEM109 protein from the RM lysate to particular hSnd2 peptides was evaluated by antibody detection and chemiluminescence. TMEM109 was found to bind either the N- or C-terminus of hSnd2, which are likely located in the cytosol in the native protein (Figure 5C,D, upper panel). When the TMEM109 peptide array was tested, solubilized hSnd2 protein was found to bind preferentially to fragments that represent the C-terminal portion of TMEM109 (Figure 5D, lower panel). Based on these peptide arrays, the TMEM109 C-terminus might likely interact with the cytosol facing N- and/or C-terminus of hSnd2.

We further verified the proximity between hSnd2 and TMEM109 under physiological conditions, i.e., in living cells using a split-luciferase system (NanoBiT) that we previously determined to be suitable for use with ER membrane proteins [70,71]. Briefly, this published work confirmed the well-known interactions of Sec61α with Sec61β, TRAPα, and Sec63, as well as between TRAPα and TRAPβ and Sec63 and Sec61β. Thus, it set the stage for addressing the putative interaction between hSnd2 and TMEM109 in living cells [71].

Here, we added the 17-kDa-large portion of the split-luciferase as a C-terminal tag to TMEM109 (TMEM109-C_L_), and the 1 kDa small portion was used to tag the N- or C-terminus of hSnd2 (hSnd2-N_S_, hSnd2-C_S_) (Appendix A). The heterologous, cytosolic Halo protein carrying the 1 kDa tag at its C-terminus (Halo-C_S_) served as a non-interacting negative control for normalization. Cells were co-transfected with two plasmids, one encoding a fusion protein carrying the larger 17 kDa tag and the second expressing the fusion construct that harbors the small 1 kDa tag. If the two fusion proteins are in close enough proximity to interact, the large and small tag reconstitute a functional luciferase that emits light upon the addition of the membrane-permeable substrate furimazine [72]. For all fusion constructs with the larger 17 kDa tag, co-transfection with the Halo-C_S_-encoding plasmid was used for normalization purposes (Appendix A). The luminescence detected in this condition, which was run in parallel to other tested co-transfections, was set to 1. Supporting the data from the co-immunoprecipitation and peptide spot arrays, the live cell protein–protein interaction assay demonstrated that the C-terminus of TMEM109 interacts with the N-terminus of hSnd2 (Figure 5E). In general, protein targeting receptors guide incoming polypeptides to a suitable translocase [8,73,74,75], hence crosslinking data showed that SRα binds to the Sec61 translocon [76]. Similarly, we found TMEM109 to interact strongly with the C-terminus of TRAPα and less efficiently with the C-terminus of Sec61α (Figure 5E). TRAPα and Sec61α are two components of the TRAP and Sec61 complex, respectively, both of which belong to the Sec61 translocon [50,77,78,79,80]. The bimolecular complementation approach is also suited to identify homo-oligomers, which we identified here for TMEM109 (Figure 5E). This finding aligns with the previous data by Venturi et al. showing that TMEM109 transiently assembles into a larger, dimeric-to-hexameric oligomer [66]. As expected, the plasmid-driven overexpression of untagged hSnd2 or TMEM109 was able to compete with the TMEM109-C_L_–hSnd2-N_S_ interaction as well as the formation of TMEM109-C_L_:–TMEM109-C_S_ oligomer (Figure 5F). To further determine the physical interaction of the human SND targeting receptor with the ER translocon, we also tested some interactions using the N-terminally tagged variant of Sec61α (Sec61α-N_L_). In comparison to the negative control Halo-C_S_, we found strong interactions of Sec61α with TRAPα-C_S_ and the N-terminus of hSnd2 using the hSnd2-N_S_ fusion protein (Figure 5G). However, Sec61α-N_L_ did not interact with TMEM109-C_S_, nor did it show the potential for oligomerization with Sec61α-C_S_ under these native conditions (Figure 5G) [81]. The results are summarized together with the previous results of the proof-of-principle study [71] in Appendix A.

Given that TMEM109 was found in the pool of proteins with reduced abundance upon the depletion of hSnd2, it could also represent a client of the SND pathway rather than an additional receptor component next to hSnd2. To exclude the possibility of TMEM109 representing a substrate of the SND pathway, in vitro synthesis and protein transport into semi-permeabilized cells was used to demonstrate the efficient membrane insertion and signal peptide cleavage of TMEM109 upon depletion of hSnd2. Considering the cleavable signal peptide, TMEM109 is likely an SRP-dependent substrate, a hypothesis that is supported by its behavior following SRα knockdown (Appendix A). In comparison to the ER of semi-permeabilized cells treated with the control siRNA, the transport of TMEM109 was improved upon depletion of hSnd2 (Figure 5H). This finding aligns well with our previously published work showing improved transport of SRP-dependent substrates upon deletion of the mammalian SND component hSnd2 [28].
Figure 5Biochemical and Cell Biological Characterization of TMEM109 as hSnd3. (**A**) Venn diagram summarizing the numbers and overlaps of candidate proteins found after siRNA-mediated hSnd2 depletion (yellow circle), siRNA-mediated hSnd2 plus Wrb double depletion (purple circle), and proteins that were previously published as hSnd2 interaction partners using co-immunoprecipitation (turquoise circle) [28]. Protein names are listed in Appendix A. The single candidate is TMEM109. Proteins from the LFQ MS analysis that showed a log2 fold value <−1 and a *p*-value < 0.01 for both siRNAs were considered as candidates. (**B**) Co-immunoprecipitation using the supernatant of CHAPS-solubilized HeLa lysates after centrifugation (Input). α-hSnd2 (left panels) or α-TMEM109 (right panels) antibodies were used for immunoprecipitation in combination with protein A/G sepharose. A sample of the supernatant after immunoprecipitation (IP) is shown as well as a sample of the first washing step (Wash) and the eluate after the acidic pH shift. Proteins of interest indicated on the right were detected by Western blotting. The original blots with molecular mass markers are shown in Appendix A. (**C**) Aplhafold structural models of TMEM109 and hSnd2. The N- and C-termini are indicated. The first 45 residues (including the 33-amino-acid-long signal peptide) were deleted from the N-terminus of the TMEM109 model (N*). Polar and charged residues of hSnd2 reminiscent of a hydrophilic vestibule are highlighted as stick models in red. The orange circles highlight the oligopeptides within the coiled-coil domain of TMEM109 (_224_EELRWRQRR_232_) and in the C-terminal domain of hSnd2 (_156_EHNEKRQRR_164_), respectively, that show sequence similarity. (**D**) Peptide spots that cover the primary structure of hSnd2 (top array) and TMEM109 (bottom array). The binding of solubilized TMEM109 or hSnd2 proteins from the RM lysate to particular spots was evaluated by antibody detection. Light orange or blue rectangles represent parts of the primary structure that encode a TMH or SP, respectively. Spots that are likely accessible in the native protein and repeatedly show a strong interaction (n = 3) are highlighted by dashed rectangles in red. (**E**) Bimolecular luminescence complementation was used to detect protein–protein interactions between TMEM109 and various partners based on relative luminescence units (RLU). TMEM109 was tagged at its C-terminus with the LargeBit component of the NanoLuc luciferase (TMEM109-C_L_). Putative interaction partners of interest were N- or C-terminally fused with the SmallBiT component of the NanoLuc luciferase (protein-N_S_ or protein-C_S_). The heterologous Halo-C_S_ fusion protein served as a non-interacting negative control for normalization and was set to 1 [71]. (**F**) The interaction between TMEM109-C_L_ and hSnd2-N_S_ (blue bars) as well as TMEM109-C_L_ and TMEM109-C_S_ (green bars) was challenged by complementation with an untagged interaction partner protein. In comparison to the empty vector (EV) control, plasmids that encode for an untagged variant of one of the interaction partners reduced the RLU. (**G**) Same as in (**E**), but Sec61α-N_L_ served as a reporter component. (**H**) In vitro transport of ^35^S-Met-labeled TMEM109. The TMEM109 precursor (pTMEM109) was synthesized in the absence (No ER) or presence of semi-permeabilized cells pretreated with control siRNA (siCtrl) or *HSND2* siRNA (si*SND2*) for 96 h before semi-permeabilization. The orange color highlights the oligopeptide _224_EELRWRQRR_232_ within the coiled-coil domain of TMEM109 that shows sequence similarity to an oligopeptide in the C-terminal domain of hSnd2 (_156_EHNEKRQRR_164_). Statistical comparison of multiple conditions was performed using one-way ANOVA followed by Dunnett’s multiple comparison post-test (**E**,**G**). Statistical comparison of the competition experiment (**F**) was based on a Student’s *t*-test comparing the EV treatment versus the corresponding hSnd2 or TMEM109 competition. *p*-values are indicated by asterisks with *p* < 0.05 (*) < 0.01 (**) < 0.001 (***).
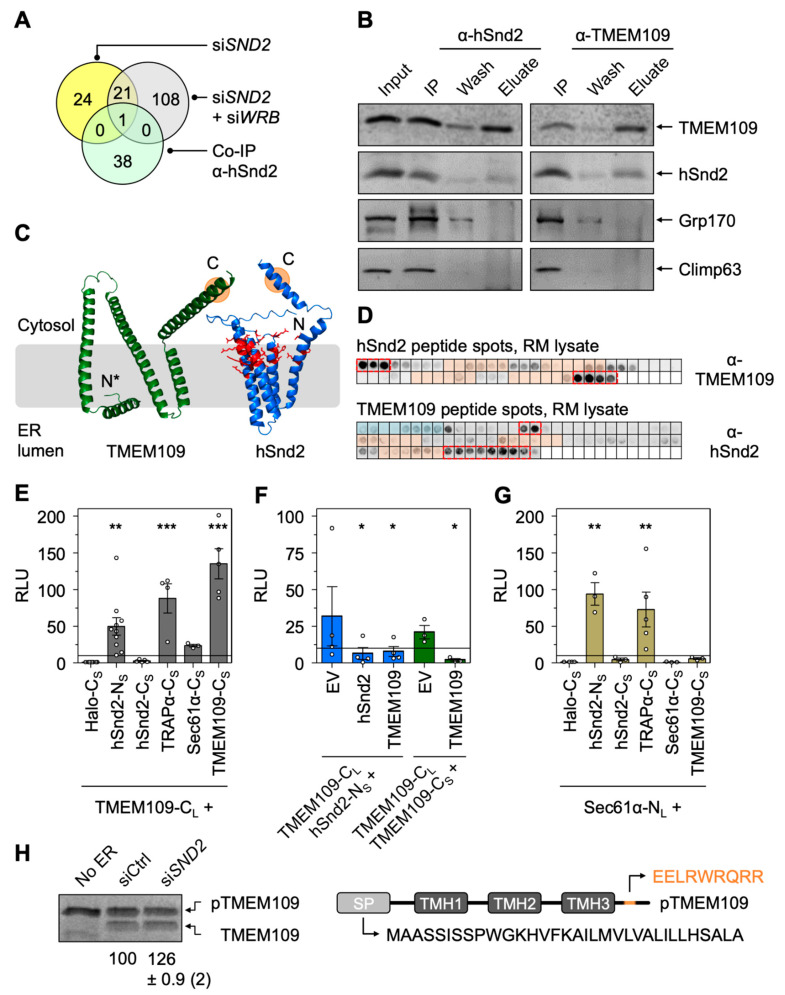



### 3.8. Depletion of TMEM109 Phenocopies the Cellular Alterations Found after Knockdown of hSnd2 

As a membrane protein of the ER that interacts with the orphan mammalian hSnd2, we wondered if TMEM109 represents a functionally relevant receptor component of the SND pathway. Upon establishing the siRNA-mediated TMEM109 silencing that was confirmed by Western blot analysis (Figure 6A–C), characteristic phenotypes occurring after the depletion of hSnd2 were analyzed. The phenotypes of interest are (i) the increased abundance of SRα/SRβ (Figure 6B,C), (ii) no apparent UPR induction based on the unchanged abundance of ER luminal chaperones BiP and Grp170 (Figure 6B,C), (iii) the improved transport of SRP-dependent substrates (preprolactin and invariant chain) as part of the compensatory response to the loss of the human SND pathway activity (Figure 6D,E), (iv) the unaltered insertion efficiency of the TRC-dependent substrate Sec61β (Figure 6D,E), and (v) the reduced transport of the partially SND-dependent substrate Cytb5 (Figure 6D–G). For all five of these indicators, the knockdown of TMEM109 clearly phenocopied previously established findings from the knockdown of hSnd2 [28]. 

To further support the idea that hSnd2 and TMEM109 could act as a functional unit of the SND pathway, the cell-free ER import of other potential clients of hSnd2 was carried out employing semi-permeabilized HeLa cells, which had or had not been treated with targeting siRNA, in combination with rabbit reticulocyte lysate, ^35^S-methionine and in vitro transcribed mRNA (Figure 6F). The established hSnd2 client Cytb5 and other OPG2-tagged chimeras of the single-spanning type III membrane proteins GypC, BCMA, and Syt1 (with apparent delta G values for membrane insertion between −2.67 and −4.1 kcal/mol) served as model proteins [42]. SRα knockdown cells were analyzed for comparison and N-glycosylation efficiency was used as a readout for ER import via SDS-PAGE and phosphorimaging. In summary, the membrane insertion of those model precursor proteins was negatively affected by the SRα, hSnd2, or TMEM109 knockdown as well as combinations thereof. However, the effects after SRα depletion were not significant for any tested substrate and for TMEM109 single depletion only in the case of Cytb5 (Figure 6G). One of the three independent TMEM109 silencing experiments had a substantially less efficient knockdown, which might explain the larger SEM and lack of significance for the tested type III membrane proteins. Interestingly, for the substrates Cytb5 and Syt1, the TMEM109 plus hSnd2 double depletion showed an additive effect further lowering the membrane insertion compared to the incomplete TMEM109 or hSnd2 single depletion (Figure 6F,G). Furthermore, analysis of the MS data also drew our attention to the ER hairpin protein REEP5, which was affected in the *SND/WRB* double silencing (Figure 2A). Considering the monotopic topology of the two REEP5 hairpins facing towards the cytosol, a carbonate extraction was used to determine its membrane insertion efficiency when in vitro synthesized in the presence of different semi-permeabilized HeLa cells [82]. In comparison to the control siRNA treatment, the depletion of either hSnd2 or TMEM109 reduced the membrane insertion of REEP5, adding this protein to the list of SND clients (Figure 6H). 

Thus, the cell-free analysis of potential SND substrates supports the conclusion from the MS analysis: namely that single-spanning type III or type IV membrane proteins with different properties (membrane protein type, positioning of the TMH, delta G) as well as the hairpin protein REEP5 can be targeted to the human ER by hSnd2 or TMEM109. These experiments also highlight the potential functional ability of TMEM109 to serve as the human hSnd3 homolog acting in concert with its interaction partner hSnd2.

## 4. Discussion

About 30% of all human proteins are imported into the ER. Importing proteins relies on cleavable or non-cleavable targeting signals in the precursor polypeptides and comprises two major stages: (i) the targeting of the nascent or fully synthesized proteins to the ER and (ii) their subsequent insertion into or translocation across the ER membrane. To accommodate the plethora of import substrates with varying primary structures and targeting signals, the import process relies on many specialized factors. Typically, cytosolic factors first recognize the signals and facilitate their ER targeting via interaction with dedicated receptors in the ER membrane and, subsequently, support client transfer for membrane insertion or translocation by the Sec61 complex or its supercomplex with the TMCO1 (Figure 7B) [3,4]. So far, four protein targeting pathways which can deliver precursor polypeptides to the ER membrane in human cells have been described, i.e., the SRP/SR-, the SGTA/TRC/WRB-, the SND-, and the PEX19/PEX3-dependent pathway [1,14,73,74]. While there are functional and structural insights available for the molecular mechanisms and client spectra of the SRP- and TRC-dependent pathways, little is known about the composition and mechanistic function of the human SND pathway.

Therefore, the dual aim of this study was to gain further insights into the components and client spectrum of the SND pathway in human cells. For this, we used the previously successfully established experimental strategy of combining the transient depletion of a central membrane component of the pathway using two different *HSND2* targeting siRNAs in parallel to a non-targeting siRNA from HeLa cells, with total cellular proteomic analysis by label-free quantitative MS analysis and differential protein abundance analysis [34,35,36,37,38]. As an internal proof of principle, as well as for comparison, similar experiments were carried out for the SRP- and TRC-dependent targeting pathways for which data from different global approaches, including ribosome profiling and component trapping data, are already available [39,40,41]. In addition, a combined hSnd2 and Wrb depletion was carried out to give insights into the client overlap between the two targeting pathways under physiological conditions, such as fast translation rates and competing precursor proteins. 

### 4.1. Depletion of hSnd2 and hSnd2 plus Wrb Identifies Multispannig Membrane Proteins and Precursors with C-Terminal Targeting Signals as Comparatively Predominant Clients of the Human SND Pathway

The proteomic analysis of the clients of hSnd2 as well as hSnd2 plus Wrb suggests that there is a clear preference of the human SND pathway for membrane proteins, including membrane proteins with cleavable N-terminal SPs (Figure 2A,B). This is consistent with previous work from various laboratories which characterized the TA protein Cytb5 and the multi-spanning membrane protein TRPC6, as well as additional TA and GPI-anchored membrane proteins, as substrates of the SND pathway in human cells [17,27,28,83,84]; and with this work identifying the hairpin protein REEP5, together with various type III membrane proteins (Figure 6F–H). During the course of this study, it also became clear that the human SND- and TRC-dependent pathways have a substantial overlap in clients, as evidenced after the simultaneous knockdown of both pathways. Furthermore, we also find that hSnd2 and Wrb clients include many multi-spanning membrane proteins, in addition to those that have relatively central or C-terminal TMHs as compared to SRα-dependent membrane proteins (Figure 2B,G). Similar observations were previously made for the SND pathway in yeast [17] and with respect to TA proteins for the GET/TRC pathways in yeast as well as human cells [16,41], where there are clear differences for the SRP pathway [39,40]. These findings are all consistent with our observation that there was very little overlap between SRα clients and the clients of the other two pathways (Figure 2A). Unexpectedly, however, we identified many more Wrb as well as Wrb and hSnd2 clients than just TA proteins, including proteins with relatively central and even N-terminal TMHs (Figure 2A,G). These findings may point towards both a more general targeting role of the TRC pathway than previously anticipated and a contribution to the congenital disorders of glycosylation in human patients with pathogenic variants of Caml, TRC35, or TRC40 [57]. Together with previous observations that small human presecretory proteins can be targeted to the ER of semi-permeabilized human cells by SRα, Wrb, and hSnd2 [29,33] and that the cytosolic quality control and TRC pathway component SGTA is cotranslationally recruited to ribosomes, which synthesize a diverse range of membrane proteins, including those with cleavable SP [58], the possibility of a more general role of the TRC pathway in ER targeting undoubtedly warrants future study.

Interestingly, there was also hardly any overlap between the previously reported PEX3-dependent clients and the substrates of any of the other protein targeting routes, SRP, SND, or TRC, with exception to the TA protein FAR1 (PEX3 and Wrb) and the type I membrane protein HLA-C (PEX3, SRα and Wrb) (Figure 2A). When human patient fibroblasts with a complete PEX3 deficiency were subjected to MS and differential protein abundance analysis [38], the PEX3 knockout negatively affected clients with a SP (56%) to an even greater extent than the SRα knockdown employed in this study (44%) and, likewise, gave an even higher enrichment of C-terminal targeting signals (32%) in clients with TMH as compared to hSnd2 or Wrb knockdown (21% and 27%, respectively) (Figure 2B,G). In contrast to hSnd2 and Wrb, PEX3 did not show any preferences for multi-spanning membrane proteins and had a lower preference for membrane proteins with central targeting signals (Figure 2D–G). Notably, however, unlike the other three ER membrane receptors, PEX3 appears to be restricted to certain areas of the ER or ER subdomains, where lipid droplets are formed and peroxisomal precursors and large cargo secretory vesicles are budding off [31,38,85]. This restricted localization may explain, by an as of yet unknown mechanism, its different client spectrum (Figure 2A and Figure 7B).

## 4.2. hSnd2 and TMEM109 form the Heterodimeric Receptor of the SND Pathway in the Human ER Membrane

Based on the previous observation that the subunits of ER membrane protein complexes such as Sec61 and TRAP are degraded after depletion of the Sec61α and TRAPβ subunits [34], respectively, we expected the elusive human Snd3 orthologue to be negatively affected after the depletion of hSnd2 from HeLa cells. This view was further substantiated here by the fact that SRβ and Caml were negatively affected in the absence of their interaction partners in the heterodimeric protein targeting receptors, SRα and Wrb, respectively (Figure 1A,B). TMEM109, which had previously been identified as a potential hSnd2 interaction partner via co-immunoprecipitation experiments, employing anti-hSnd2 antibodies [28], was observed amongst the negatively affected proteins after the depletion of hSnd2 as well as hSnd2 and Wrb (Figure 2A, Appendix A) and confirmed as a robust hSnd2 interaction partner by co-immunoprecipitation using anti-TMEM109 antibodies (Figure 5B). In light of the observation that TMEM109 was not an hSnd2 substrate under the conditions of cell-free transport into the ER of semi-permeabilized HeLa cells (Figure 5H), these results were taken as a first indication that TMEM109 may, indeed, represent the hSnd3. Subsequently, this idea was further substantiated by peptide array experiments (Figure 5E,G) as well as by the split-luciferase analyses in intact HeLa cells. In full agreement with the data from the reciprocal co-immunoprecipitations outlined above and from peptide spot arrays (Figure 5D), the live cell protein–protein interaction assay demonstrated that the C-terminus of TMEM109 interacts with the N-terminus of hSnd2 (Figure 5F). Using the same approach, we found TMEM109 to interact with the C-terminus of TRAPα and the C-terminus of Sec61α (Figure 5E,G), which is also consistent with BioGRID data (see below). Notably, TRAPα and Sec61α are components of the Sec61 translocon, consistent with the idea that the human SND pathway can target proteins to the Sec61 translocon [29,33].

Last but not least, various functional assays support the notion that the hSnd2 interaction partner TMEM109 is the hitherto elusive hSnd3. The knockdown of TMEM109 phenocopied all previously established findings from the knockdown of hSnd2 [28], i.e., the increased abundance of SRα/SRβ (Figure 6B,C) in combination with the absence of UPR induction (Figure 6B,C), the improved in vitro transport of SRP-dependent substrates using semi-permeabilized HeLa cells (Figure 6D,E), the unaltered insertion efficiency of the TRC-dependent substrate Sec61β (Figure 6D,E), and the reduced transport of the previously identified SND substrate Cytb5 as well as the hairpin model protein REEP5 (Figure 6D–H). Likewise, the membrane insertions of three single-spanning type III membrane proteins, GypC, BCMA, and Syt1, were each negatively affected by the knockdown of SRα, hSnd2, or TMEM109, as well as combinations thereof, under the same in vitro conditions. However, the effects of SRα well as TMEM109 depletion were mild (25% reduction) and not statistically significant, while the effects of hSnd2 depletion were both stronger and significant, with an even more pronounced defect observed after the simultaneous knockdown of hSnd2 and TMEM109 (Figure 6F,G). Therefore, it appears safe to conclude that TMEM109 represents the functional homolog of yeast Snd3, i.e., hSnd3. Notably, the efficient ER insertion of the type III membrane proteins GypC and BCMA is dependent on the concerted actions of the EMC and Sec61 complex [42]. Thus, the human SND pathway can apparently target precursors to the EMC as well as the Sec61 channel, thereby supporting the insertion of the broad spectrum of client proteins that rely on different membrane insertases (Figure 7B). Considering the cellular abundance of the individual targeting receptor components, the corresponding partner proteins show similar steady-state levels (Figure 7A “abundance”): SRα/SRβ (249/173 nM), Wrb/Caml (4/5 nM), hSnd2/TMEM109 (81/49 nM) [30].

Of note, TMEM109 is also termed Mitsugumin 23, or MG23, and was found in nuclear and ER as well as SR membranes, where it can form a large bowl-shaped homooligomer that behaves as a voltage-dependent K^+^ and Ca^2+^ channel after reconstitution into planar lipid bilayers [66,86]. Together with BRI3BP, which was originally predicted as a mitochondrial outer membrane protein with four transmembrane domains, it forms the Mitsugumin 23 protein family (Appendix A). However, according to the most advanced prediction tools, BRI3BP may in fact share its general architecture with TMEM109, i.e., an N-terminal SP [87], three transmembrane domains, and a C-terminal coiled-coil domain (Appendix A). The latter two features are reminiscent of EMC3 [69], although the coiled-coil domain of BRI3BP lacks the short oligopeptide (_224_EELRWRQRR_232_) with homology to hSnd2, which may be involved in the interaction of TMEM109 with hSnd2. Notably, according to BioGRID, Sec61β, as well as various interaction partners of the Sec61 complex, including Sec62, TRAPα and Ribophorins I and II, are among the 88 and 109 interactors of TMEM109 and BRI3BP, respectively, that were identified by Proximity Label-MS (https://thebiogrid.org; accessed 12 September 2022). We therefore speculate that both BRI3BP and TMEM109 may be associated with the Sec61 translocon, suggesting that BRI3BP is probably located in the ER membrane and might even be able to substitute for TMEM109 in the SND pathway. Consistent with this view, BRI3BP was upregulated after Wrb depletion (Appendix A), although it was not quantified in the other MS experiments (Appendix A). Whilst we did not identify BRI3BP as an interaction partner of hSnd2, the possibility that BRI3BP and TMEM109 are both paralogs with respect to ER protein targeting should be addressed in future work.

## 4.3. A working Hypothesis for the Human SND Pathway

Although the two original questions relating to additional components and the client spectrum of the human SND pathway were answered here, some open questions remain. The first concerns BRI3BP (see above), the second addresses the cytosolic component(s) acting upstream of the hSnd2/hSnd3 receptor, and the third relates to the possible mechanism of hSnd2/hSnd3. The four protein targeting pathways, which can deliver precursor polypeptides to the ER membrane, appear to share a common principle, namely that they comprise a cytosolic component or complex, which recognizes SPs and TMHs in newly synthesized ER import substrates (SRP, TRC35/Ubl4A/Bag6/TRC40, Snd1 in yeast, PEX19), plus a heterodimeric membrane receptor for the cytosolic SP- and TMH-recognition component (SRα/β, Wrb/Caml, hSnd2/hSnd3, PEX3 maybe together with PEX16) [1,14,73,74]. Notably, however, a functional human homolog of yeast Snd1 which was shown to interact with ribosomes has not been identified, leaving open the question as to which component(s) act(s) upstream of hSnd2/hSnd3. The cytosolic proteins Calmodulin [88,89], Hsc70 [90], NAC [91], N-acetyl transferase (NAT) [92], RAC [93], SGTA [94], and Ubiquilins [95,96] all accept nascent polypeptide chains or fully synthesized proteins from ribosomes and have previously been implicated in intracellular protein transport in one way or another [8,58,90,94,97,98]. Except for NAT, RAC, and NAC, which appear to antagonize SRP, all the other proteins act in combination with one or more of SRP, TRC, and PEX19 to mediate a molecular triage process that decides between protein targeting and protein degradation, interact with amphipathic helices, such as SPs and TMHs, and are therefore potential candidates to serve as hSnd1 in the hSND pathway. Notably, Ubiquilin 2 was found to be co-immunoprecipitated together with hSnd2 [28]. Of further note, the cellular concentration of all cytosolic components of the TRC pathway exceed the concentration of the Wrb/Caml receptor by one to two orders of magnitude (177–549 versus 4–5 nM) [30]. Thus, one or more of these components may also work upstream of hSnd2/hSnd3, which would be consistent with the overlapping client spectrum between the two pathways.

Another intriguing consideration relates to the working model of the SND pathway and its hSnd2/TMEM109 receptor and is based on the broad spectrum of clients that it transports (Figure 7B). These include different types of membrane protein clients, such as co-translationally inserted type III and post-translationally inserted TA proteins (Figure 2 and Figure 6). Similar to the yeast Snd2 protein that shows a hydrophilic vestibule reaching into the presumed membrane environment [99], the predicted structure of the human hSnd2 also shows a ring of hydrophilic residues that align at the cytosolic leaflet of the ER membrane (Figure 5C). Thus, depending on the incoming precursor polypeptide, the dimeric hSnd2/hSnd3 receptor might also have an insertase function, equivalent to that of the Wrb/Caml receptor or EMC insertase, which operates through the local distortion or thinning of the membrane [74]. Similarly, TMEM109 might represent a moonlighting protein that acts either as a cation channel upon homooligomerization or as a receptor component when associated with hSnd2 (Figure 7B). This behavior is reminiscent of TMCO1, which was described as part of the ER protein insertase handling polytopic membrane proteins as well as acting as a cation channel of the ER membrane to prevent ER calcium overload upon its tetramerization [23,100].

## 5. Conclusions

Four pathways target precursor proteins to the human ER. TMEM109 was characterized as a subunit of the receptor for the SRP-independent/SND pathway and represents the hitherto unknown hSnd3, the hSnd2 partner in the ER membrane. Multipass membrane proteins were identified as predominant clients of the human SND pathway. Unlike the SRP-dependent pathway, the TRC-dependent pathway has an overlapping client spectrum with the SND pathway, which explains why the two pathways can substitute for each other. SND- and TRC-dependent pathways target precursors with central or C-terminal targeting signals.

## Figures and Tables

**Figure 1 cells-11-02925-f001:**
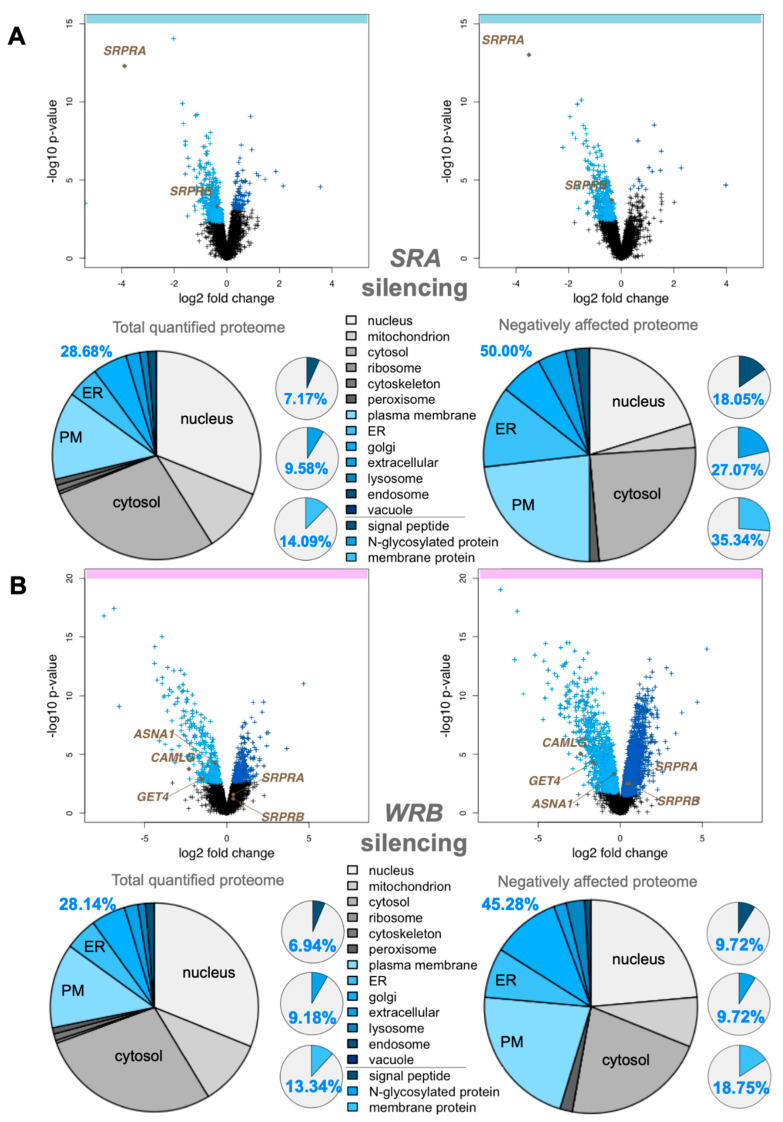
Identification of SRα, and Wrb Clients by MS after SRA or WRB Silencing in HeLa Cells. (**A**,**B**) The experimental strategy was as follows: siRNA-mediated gene silencing using two different siRNAs for each target and one non-targeting (control) siRNA with six/three replicates for each siRNA in two/one independent experiments, respectively; label-free quantitative proteomic analysis; and differential protein abundance analysis to identify negatively affected proteins (i.e., clients) and positively affected proteins (i.e., compensatory mechanisms). Original data are shown in Appendix A. Knockdown efficiencies were evaluated by Western blot (Appendix A). Differentially affected proteins were characterized by the mean difference of their intensities plotted against the respective permutation-false-discovery-rate-adjusted *p*-values in volcano plots (n = 2 in the case of SRα depletion (**A**), n = 1 in the case of Wrb depletion (**B**)). The results for a single siRNA are shown in each case. Subunits of the SRP receptor (**A**) and Wrb interaction partners (**B**) are indicated, respectively. For the validation of clients, protein annotations of signal peptides, membrane location, and N-glycosylation in humans were extracted from UniProtKB and used to determine the enrichment of Gene Ontology (GO) annotations among the secondarily affected proteins. The colors of GO annotations (large pies) and the three others (small pies) are indicated in the Figure. Statistical analysis was carried out as described in Materials and Methods. Some pie sections were additionally defined by labeling for better orientation, such as ER, endoplasmic reticulum; PM, plasma membrane.

**Figure 2 cells-11-02925-f002:**
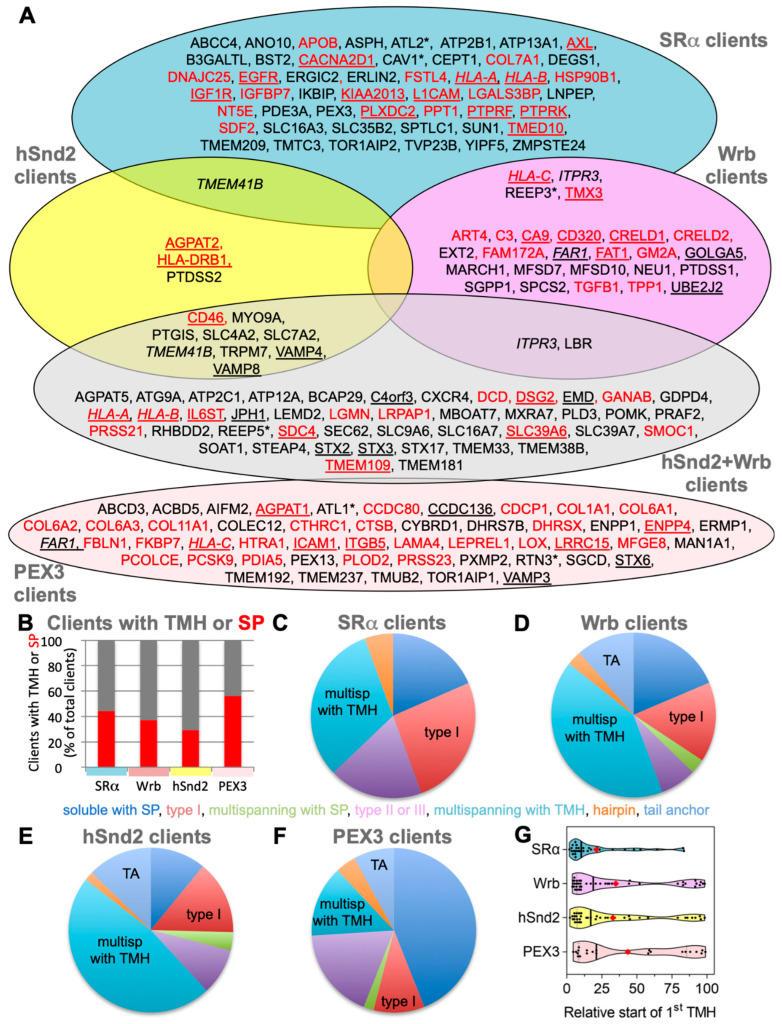
Venn Diagrams for and Distinguishing Features of SRα, Wrb, and hSnd2 Clients. (**A**) The clients of various components for protein targeting to the human ER, which were determined here by quantitative MS and differential protein abundance analysis following depletion of the respective component in HeLa cell for 96 h, are shown in a Venn diagram. Clients are defined as such by the presence of either an SP or at least one TMH. Shown with their gene names, clients with SPs are shown in red, SP containing membrane proteins are underlined, clients with TMH are shown in black, TA membrane proteins are underlined, hairpin proteins are indicated by asterisks, and italics highlight clients of two targeting components which could not be properly fitted into the Venn diagram and are named twice. The data for PEX3 were taken from Zimmermann et al. [38]. (**B**) To characterize the clients of the various targeting components, the percentage of SP (red) and TMH (grey) containing clients was calculated; the total number of clients (n) was 54 for SRα, 70 for Wrb, 55 for hSnd2, and 50 for PEX3. (**C**–**F**) The different client types were plotted in order to show their relative distribution (colors are given in the Figure and are plotted clockwise starting from the twelve o’clock position in each cake). Original data are shown in Appendix A. (**G**) The TMH containing clients were assembled in a violin plot in order to show the location of the relative start of the first TMHs in the clients (i.e., position of central amino acid residue of TMH in % of client amino acid residues). The violin plot shows lower and upper quartiles, the median and the single data points; the red diamond identifies the average, i.e., 21+/−4.3%, 35+/−6%, 32.8+/−5.2% and 35+/−6%, and 43.5+/−8. The following distributions of clients were observed between the relative start lines between 0 and 25 (defined as more N-terminal), >25 and <75 (defined as central), and 75 and 100 (defined as more C-terminal): 77, 17, and 7% for SRα; 52, 20, and 27% for Wrb; 59, 21, and 21% for hSnd2; 55, 14, and 32% for PEX3 (Appendix A). Some pie sections were additionally defined by labeling for better orientation, such as multisp, multispanning membrane proteins; TA, tail anchored membrane proteins type I, type I membrane protein; SP, signal peptide; TMH, transmembrane helix.

**Figure 3 cells-11-02925-f003:**
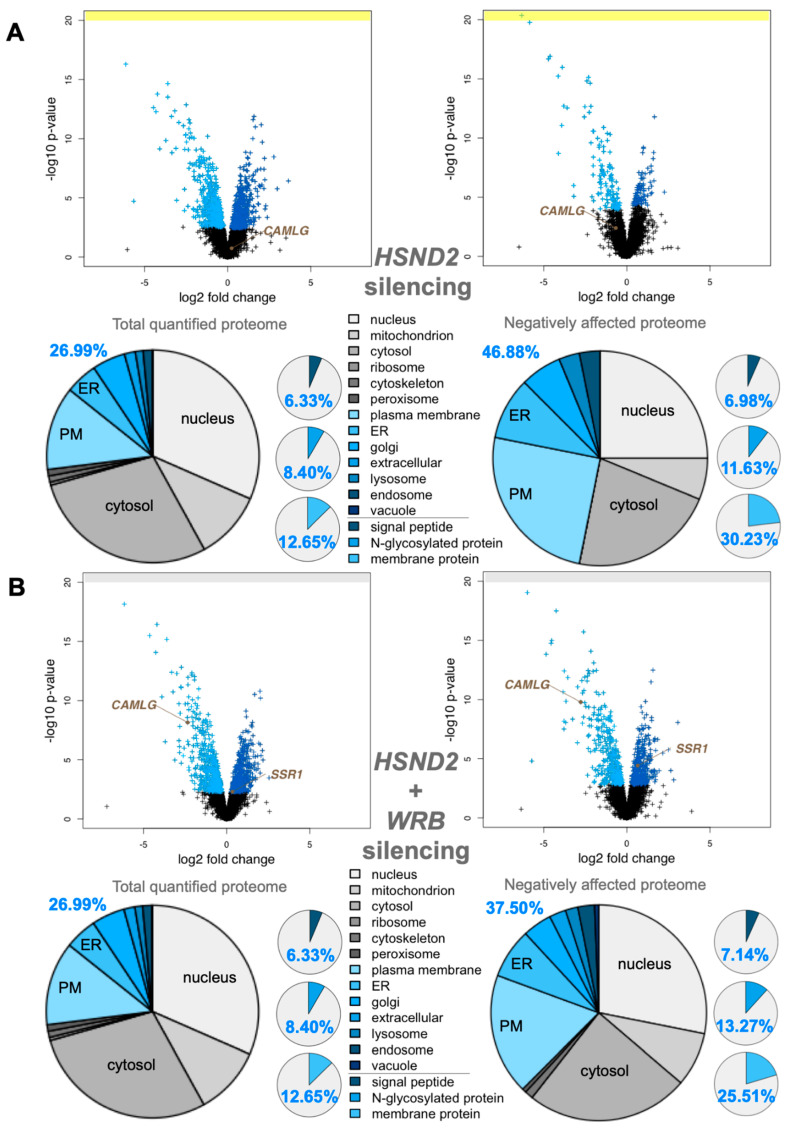
Identification of hSnd2 Clients by MS after *HSND2* or Simultaneous *HSND2* plus *WRB* Silencing in HeLa Cells. (**A**,**B**) The experimental strategy was as follows: siRNA-mediated gene silencing using two different siRNAs for each target and one non-targeting (control) siRNA with three replicates for each siRNA or combination of siRNAs in one experiment; label-free quantitative proteomic analysis; and differential protein abundance analysis to identify negatively affected proteins (i.e., clients) and positively affected proteins (i.e., compensatory mechanisms). Original data are shown in Appendix A. Knockdown efficiencies were evaluated by Western blot (Appendix A). Differentially affected proteins were characterized by the mean difference of their intensities plotted against the respective permutation false discovery rate-adjusted *p*-values in volcano plots (n = 1). The results for a single siRNA are shown in each case. The Wrb interaction partner Caml (*CAMLG* in (**A**,**B**)) and TRAPα (*SSR1* in (**B**)), respectively, are indicated. For validation of clients, protein annotations of signal peptides, membrane location, and N-glycosylation in humans were extracted from UniProtKB and used to determine the enrichment of Gene Ontology (GO) annotations among the secondarily affected proteins. The colors of GO annotations (large pies) and the three others (small pies) are indicated in the Figure. Statistical analysis was carried out as described in Materials and Methods. Some pie sections were additionally defined by labeling for better orientation, such as ER, endoplasmic reticulum; PM, plasma membrane.

**Figure 4 cells-11-02925-f004:**
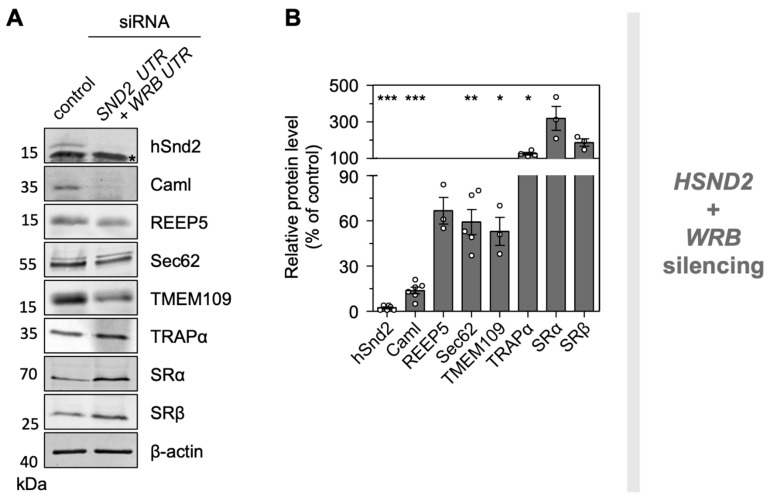
Western Blots for Validation of Selected Clients. (**A**,**B**) The experimental strategy was as follows: siRNA-mediated gene silencing using one siRNA for each target and one non-targeting (control) siRNA, respectively, in at least three independent experiments; knockdown efficiencies as well as a set of putative clients and possible compensatory proteins were evaluated by Western blots (**A**). (**B**) Quantification of the relative protein abundance based on Western blot analysis upon the indicated silencing. Statistical comparison was based on a Student’s *t*-test comparing against the control siRNA treatment that was normalized to 100% for each tested protein. *p*-values are indicated by asterisks with *p* < 0.05 (*) < 0.01 (**) < 0.001 (***). The open circles represent the individual data points. The selection of clients and compensatory components was purely based on the availability of reliable antibodies. For technical reasons, Caml was used as a proxy for Wrb. The star refers to a cross-reactivity of the anti-hSnd2 antibody.

**Figure 6 cells-11-02925-f006:**
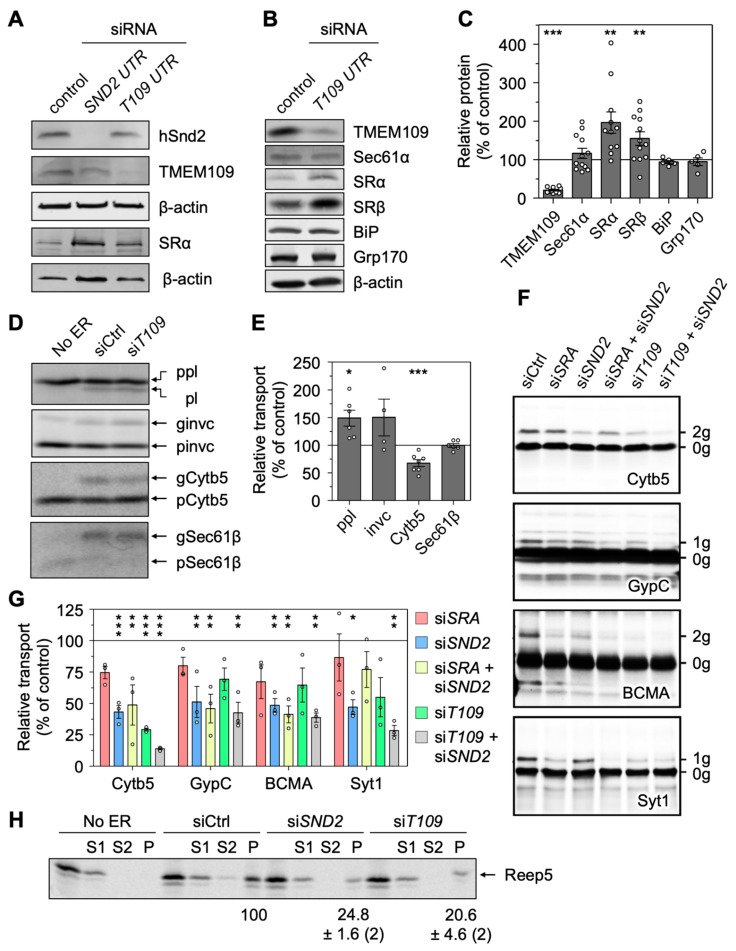
Functional Characterization of TMEM109 as hSnd3. (**A**,**B**) Western blot analysis of the indicated proteins upon treatment of HeLa cells with siRNAs targeting *HSND2* (*SND2 UTR*) or *TMEM109* (*T109 UTR*). An untargeted control siRNA was used for comparison. Cells were harvested 96 h (**A**) or 72 h (**B**) post-silencing. The original blots with molecular mass markers are shown in Appendix A. (**C**) Quantification of the relative protein abundance based on Western blot analysis upon silencing TMEM109 for 72 h. (**D**) In vitro transport of the ^35^S-Met-labeled model substrates preprolactin (ppl), invariant chain (invc), Cytochrome b5 (Cytb5), or Sec61β. Precursors are indicated by a ‘p’, and the imported, glycosylated proteins by a ‘g’. Upon transport and cleavage of the signal peptide, the indicated prolactin form is visible (pl). Precursors were synthesized in the absence (No ER) or presence of semi-permeabilized cells pretreated with control siRNA (siCtrl) or *TMEM109* siRNA (si*T109*) for 72 h before semi-permeabilization. (**E**) Quantification of the relative transport efficiency for multiple repeats of the substrates tested in (**D**). For each repeat, the substrate transport in control siRNA-treated ER fractions was set to 100%. (**F**) In vitro transport after 72 h of different siRNA treatments targeting SRα, hSnd2, TMEM109, or combinations thereof. Following in vitro translation of the OPG2-tagged model substrates Cytb5, glycophorin C (GypC), tumor necrosis factor receptor superfamily member 17 (BCMA, containing an artificially inserted N-glycosylation site), and synaptotagmin 1 (Syt1), the reactions were solubilized using Triton X−100. Total radiolabeled products (i.e., membrane-associated and non-targeted nascent chains) were recovered by immunoprecipitation via the OPG2-tag and analyzed by SDS-PAGE and phosphorimaging. The N-glycosylation of lumenal domains, previously confirmed by treatment with endoglycosidase H [42,43], indicates successful membrane translocation/insertion (1 g or 2 g) in contrast to non-inserted precursor proteins (0 g). (**G**) Relative membrane insertion efficiencies were calculated using the ratio of N-glycosylated protein to non-glycosylated protein, relative to the control siRNA-treated cells (set to 100% insertion efficiency). Quantifications are given as means ± SEM for independent insertion experiments from separate siRNA treatments performed in triplicate (n = 3). (**H**) Carbonate extraction of the ^35^S-Met labeled model substrate Reep5. The total translation reaction (T) was centrifuged through a sucrose cushion, and the pellet was resuspended in alkaline carbonate solution before a second centrifugation step through an alkaline sucrose cushion occurred. Samples of supernatants after the first (S1) and second (S2) centrifugation as well as the final pellet (*p*) that includes the membrane-integrated Reep5 are shown. Statistical comparison of multiple conditions was performed using a one-way ANOVA followed by Dunnett’s multiple comparison post-test (**G**). Statistical comparison of *TMEM109* silencing (**C**,**E**) was based on a Student’s *t*-test comparing against the control siRNA treatment that was normalized to 100% for each tested protein (**C**) or substrate (**E**). *p*-values are indicated by asterisks with *p* < 0.05 (*) < 0.01 (**) < 0.001 (***).

**Figure 7 cells-11-02925-f007:**
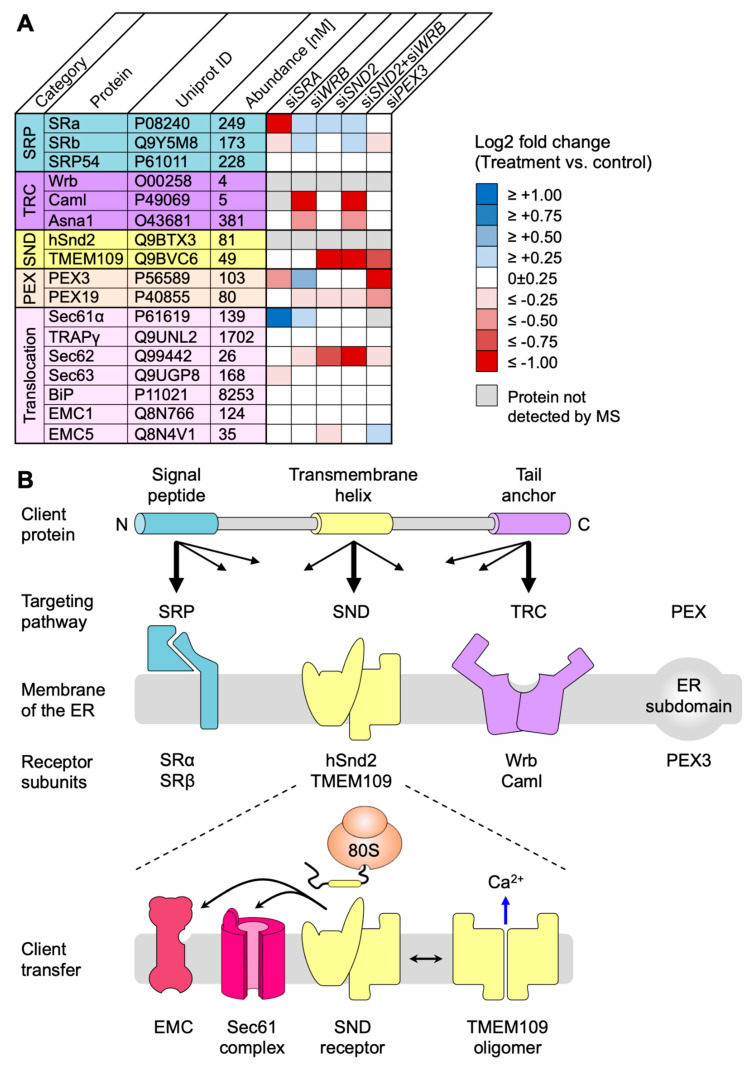
Interconnectivity of ER Protein Targeting Pathways and a Working Model for the Function of hSnd2/hSnd3 Receptor in ER Membrane. (**A**) Heatmap summarizing the log2 fold changes of protein levels upon the siRNA-mediated gene silencing according to Appendix A. Proteins of interest together with their Uniprot ID and cellular abundance under steady-state conditions (taken from Hein et al. [30]) are consolidated in differently colored categories. Grey boxes report proteins that were not detected by the LFQ mass spectrometry in certain conditions. (**B**) Summary of the four targeting pathways SRP, SND, TRC, and PEX with regard to the ER membrane receptor components and their preferential client proteins based on the targeting signal or targeting to an ER subdomain. Further details can be found in the text.

**Table 1 cells-11-02925-t001:** siRNAs.

Name	Target Sequence	Source	Concentration (nM)	Time (h)
*HSND2*-UTR siRNA #2	CTCTATAGGGTCGTTGAATAA	Qiagen	20	96
*HSND2* siRNA #3	AAGGGCAAAGTGGGCACGAGA	Qiagen	20	96
*SRA*-UTR siRNA #3	CACCAGAGCTTTGCTAATAAT	Qiagen	15	96
*SRA*-UTR siRNA #6	CAGAGAAATAAGTAATTTATA	Qiagen	15	96
*TMEM109*-UTR #1	CAGGTTTGATGTGGAATCACA	Qiagen	20	72
*TMEM109-UTR #2*	CACCGCCAGTGTCATACCAAA	Qiagen	20	72
*WRB*-UTR siRNA #3	TGACACGTATGTACTAGTGAA	Qiagen	20	96
*WRB* siRNA #4	CACAGTCAACATGATGGACGA	Qiagen	20	96

**Table 2 cells-11-02925-t002:** PCR primers used to create plasmids for live cell protein–protein interaction.

Construct	Fwd Primer (5′-3′)	Rev Primer (5′-3′)
hSnd2-C_S_	TAATACGACTCACTATAGG	TCCCGGTG*CTCGAG*TAtaaccgcttcatctg
hSnd2-N_S_	GTGGCCCTC*GAATTC*GAGGAGATCTGCCG	AGGGCCAC*TCTAGA*ttataaccgcttcatc
TMEM109-C_L_	TAATACGACTCACTATAGG	GAGGGCCAC*GAGCTC*Tctcctcctccacac
TMEM109-C_S_	TAATACGACTCACTATAGG	GAGGGCCAC*GAGCTC*Tctcctcctccacac
Sec61α-C_S_	AAGTG*GCTAGC*atggcaatcaaatttc	CCGT*GAATTC*GTgaagagcagggccc
Sec61α-N_L_	AGCG*CTCGAG*Gatggcaatcaaatttc	GCC*GCTAGC***TCA**gaagagcagggccc
TRAPα-C_S_	TAATACGACTCACTATAGG	GGGCCAC*GAGCTC*Cctcatcagatcccac

Sequences matching cDNA are written in uncapitalized letters, overhangs in capital letters, restriction sites in italic, and inserted stop codons in bold.

**Table 3 cells-11-02925-t003:** PCR primers used to create transcription templates.

Recombinant cDNA	Vector	Species	Forward Primer (5′-3′)	Reverse Primer (5′-3′)	RNA Polymerase
BCMA-Y13T-OPG2	pCMV3	Human	CGCAAATGGGCGGTAGGCGTG	TAGAAGGCACAGTCGAGG	T7
Cytb5OPG2	pcDNA3	Human	CGCAAATGGGCGGTAGGCGTG	TAGAAGGCACAGTCGAGG	T7
GypCOPG2	pGEM4	Human	GTGGATAACCGTATTACCGCC	CTCTGACGGCAGTTTACGAG	T7
Syt1OPG2	pcDNA3	Rat	CGCAAATGGGCGGTAGGCGTG	TAGAAGGCACAGTCGAGG	T7

## Data Availability

The MS data reported in this study have been deposited to the ProteomeXchange Consortium via the PRIDE partner repository with the dataset identifiers: PXD008178, PXD011993, and PXD012078 (http://www.proteomexchange.org; accessed on 12 September 2022). A description of the datasets is given in Appendix A. Any additional information required to analyze the data reported in this paper is available from the corresponding authors upon request.

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
