# Peer review of "Proteomics Identifies Substrates and a Novel Component in hSnd2-Dependent ER Protein Targeting"

_cells, 2022, doi:10.3390/cells11182925_

Round 1

Reviewer 1 Report

One of the main requirements for maintaining organization and order within the cell is the correct localization of proteins. Approximately one-third of all proteins are originally destined for the eukaryotic endoplasmic reticulum (ER).  It is known that the majority of these proteins are delivered by the signal recognition particle (SRP). The authors Tirincsi et al present a work to identify additional components and characterize the client spectrum of the human SND (or SRP-independent targeting) pathway. Authors characterized TMEM109 as a subunit of the receptor for the SRP-independent/SND-pathway which represents the hitherto unknown hSnd3, the hSnd2 partner in the ER membrane.

The material is written consistently and logically, with a large number of statistically significant experimental results and a comparative analysis with literature data.

I would recommend this review for publication in Cells with some minor edits.

1. The titles are different in the manuscript and the supplementary materials (“Proteomics identifies substrates and a novel component in hSnd2-dependent ER protein targeting” and “Proteomics identifies substrates and a putative novel component in hSnd2-dependent protein targeting to the human ER”). Please choose the correct one.

2. It might be worth including also a review about membrane protein insertion to ER, as in the article Shao S. and Hegde R.S. (doi: 10.1146/annurev-cellbio-092910-154125).

3. Why is TMEM109 marked in bold (Line 412)?

Author Response

We are grateful to the reviewers for their positive view on our manuscript and for their suggestions for improvement. We respond to the reviewer´s comments in detail below and provide a marked-up revised manuscript to highlight the changes. Notably, additional changes include the shift of the Materials and Methods section from the end (4.) to the front (2.), to meet journal style, which changed the numbering of sections and tables as well as the order of references.

  1. The title of the supplement was changed.
  2. We prefer to stick to the more timely reviews on the subject by O´Keefe et al. (reference 3) and Hegde and Keenan (reference 4), if that is okay.
  3. The bold letters of TMEM109 were changed.

Reviewer 2 Report

The manuscript titled “Proteomics identifies substrates and a novel component in hSnd2-dependent ER protein targetingby Tirincsi et al., builds on the earlier work of Aviram et al., 2016. The work involves extensive mass spectrometry-based experiments to identify new clients for the hSnd2 assisted protein targeting to ER. The work is of larger interest to the field of protein homeostasis and quality control as it identifies many potential substrates of hSnd2 and also highlights the overlap with the other known ER insertases. However, the authors should look at the following concerns.

A.    The authors claim that TMEM109 is an essential component of Snd pathway. But no cell-base microscopy assays such as localization to the ER membrane and its colocalization with the known hSnd2 is not provided. Similar colocalization experiments with the clients would be a good add on.  

B.    The authors claim that hSnd2 and TMEM109 interact, however no immunofluorescence or interaction with full length purified proteins is not provided. The authors do provide the split-luciferase assay, which needs to further strengthened.

C.     The authors state that the knockdown efficiency is not very high and hence leads to high standard deviation between experiments (line 696). Isn’t it better to use siRNA pool rather than using single siRNA for knockdown experiments?

Minor comments:

1.     The pie diagrams in figure 1 A and B, figure 2C-F, figure 3A and B can be improved by using distinct colors or using patterns.

2.     Figure 1A represents to the mass spec data after 96 hours of siRNA treatment. The authors should provide a knockdown efficiency at different time points which gives a clear picture of how the siRNA as working?

3.     In figure 2G, the authors provide a very informative plot of the TMH position of different clients with respect to the pathway. Can they do a similar analysis for overlapping clients?

4.     Figure2B, the “clients with TMH/SP” is misleading and generally indicates a ratio. The authors instead provide a simple colored box legend.

5.     Line 301-302, the authors state “thus, similar to SRP/SR, 48% of the observed Wrb clients have N-terminal SP and 57% of the remaining membrane protein clients have rather C-terminal TMHs” does the overall number comes to 100%? If not, are there overlapping clients?

6.     Figure 4A: The western blot for wrb is missing, also add the molecular weight markings, for the same the original full blots are not provided.

7.     Figure 4B: As the difference in protein levels varies from as small as 0% to as high as 350%, the authors may provide inset (zoomed-in) for hsnd2, CamI, REEP5, Sec62 and TMEM109.

8.     Original blots for figure 5B are missing.

Author Response

We are grateful to the reviewers for their positive view on our manuscript and for their suggestions for improvement. We respond to the reviewer´s comments in detail below and provide a marked-up revised manuscript to highlight the changes. Notably, additional changes include the shift of the Materials and Methods section from the end (4.) to the front (2.), to meet journal style, which changed the numbering of sections and tables as well as the order of references.

  1. We have low resolution cell-base immunofluorescence microscopy images, which localize both hSnd2 and TMEM109 to the ER and would be happy to add them to the supplement (please see Figure below). Both show the diffuse cytoplasmic staining, which is typical for ER proteins.

However, we are convinced that the NanoBiT data are more to the point and, therefore, strengthened them as suggested by the reviewer (see answer to B).

Furthermore, we are not aware of any successful IF microscopic attempts in the literature to colocalize ER targeting components and their clients under physiological conditions and, therefore, refrained from trying it.

  1. We are demonstrating the interaction between hSnd2 and TMEM109 both in vitro and in cells with three independent experimental strategies, co-IPs of solubilized native proteins, peptide arrays that are probed with solubilized native proteins, and NanoBiT in cells. We have added additional NanoBiT data, which strengthen the validity of the assay, to the supplement (Figure S11).
  2. In general, using two different siRNAs for the same target mRNAs separately is a generally accepted strategy for minimizing unspecific effects. In our case it additionally increases the n. Furthermore, we wanted to be able to compare the new data to the previous results of similar experiments for various ER protein import components and, therefore, used the established experimental conditions with two different siRNAs (references 34-38).

  1. The pie diagrams in Figures 1-3 were improved by defining some sections in order to make orientation easier and highlight differences.
  2. We have observed for several ER membrane proteins that the respective gene silencing by siRNA typically takes less than 24 hours to deplete HeLa cells of the mRNA and that it takes up to 96 hours to efficiently deplete the cells of the protein (references 29,34,63). Furthermore, we observed that even for essential proteins like Sec61a cell growth and viability are not negatively affected during these 96 hours to a significant degree (TMEM109 silencing was found to be an exceptional case, therefore, the respective experiments were terminated after 72 h). To demonstrate the latter in every single experiment, cells were routinely analysed after 96 h of gene silencing using the Countess Automated Cell Counter in all experiments, which is described in M&M.
  3. We did the analysis also for the overlap of hSnd2 and Wrb clients as requested and included it into Figure S4.
  4. Figure 2B was improved by giving a better definition of what is illustrated.
  5. Based on Tables S4 and S5, 13 of the Wrb clients have SP, 14 have TMH and the TMHs of 8 of those 14 or 57% have non-N-terminal or more C-terminal TMHs. We tried to improve the explanation in the revised manuscript.
  6. Figure 4A: The molecular mass markings and original blots were included.
  7. Figure 4B was changed, except that we felt that a split y-axis instead of an insert looks better. Notably, for economical reasons we used Caml as proxy for Wrb, which we explain in the new legend.
  8. The original blots for Figures 5B and 6, A and B were included.
